# A deep learning system for detecting diabetic retinopathy across the disease spectrum

Ling Dai[1,2,3,9], Liang Wu [2,9], Huating Li [2,9], Chun Cai [2,9], Qiang Wu[4,9], Hongyu Kong [4], Ruhan Liu [1,3], Xiangning Wang[4], Xuhong Hou[2], Yuexing Liu[2], Xiaoxue Long [2], Yang Wen [1,3], Lina Lu[5], Yaxin Shen [1,3], Yan Chen[4], Dinggang Shen [6,7], Xiaokang Yang[8], Haidong Zou [5✉], Bin Sheng [1,3✉] & Weiping Jia [2✉]

Retinal screening contributes to early detection of diabetic retinopathy and timely treatment. To facilitate the screening process, we develop a deep learning system, named DeepDR, that can detect early-to-late stages of diabetic retinopathy. DeepDR is trained for real-time image quality assessment, lesion detection and grading using 466,247 fundus images from 121,342 patients with diabetes. Evaluation is performed on a local dataset with 200,136 fundus images from 52,004 patients and three external datasets with a total of 209,322 images. The area under the receiver operating characteristic curves for detecting microaneurysms, cotton-wool spots, hard exudates and hemorrhages are 0.901, 0.941, 0.954 and 0.967, respectively. The grading of diabetic retinopathy as mild, moderate, severe and proliferative achieves area under the curves of 0.943, 0.955, 0.960 and 0.972, respectively. In external validations, the area under the curves for grading range from 0.916 to 0.970, which further supports the system is efficient for diabetic retinopathy grading.

[1] Department of Computer Science and Engineering, Shanghai Jiao Tong University, Shanghai 200240, China. [2] Department of Endocrinology and Metabolism, Shanghai Jiao Tong University Affiliated Sixth People's Hospital, Shanghai Diabetes Institute, Shanghai Clinical Center for Diabetes, Shanghai 200233, China. [3] MoE Key Lab of Artificial Intelligence, Artificial Intelligence Institute, Shanghai Jiao Tong University, Shanghai 200240, China. [4] Department of Ophthalmology, Shanghai Jiao Tong University Affiliated Sixth People's Hospital, Shanghai 200233, China. [5] Department of Ophthalmology, Shanghai General Hospital, Shanghai Jiao Tong University, Shanghai Eye Diseases Prevention and Treatment Center, Shanghai Eye Hospital, Shanghai Engineering Center for Precise Diagnosis and Treatment of Eye Diseases, Shanghai 200040, China. [6] School of Biomedical Engineering, Shanghai Tech University, Shanghai, China. [7] Shanghai United Imaging Intelligence Co., Ltd., Shanghai, China. [8] Shanghai Institute for Advanced Communication and Data Science, Shanghai Key Laboratory of Digital Media Processing and Transmission, Shanghai Jiao Tong University, Shanghai 200240, China. [9]These authors contributed equally: Ling Dai, Liang Wu, Huating Li, Chun Cai, Qiang Wu. ✉email: zouhaidong@sjtu.edu.cn; shengbin@cs.sjtu.edu.cn; wpjia@sjtu.edu.cn

It is estimated that approximately 600 million people will have diabetes by 2040, with one-third expected to have diabetic retinopathy (DR)—the leading cause of vision loss in working-age adults worldwide[1]. Mild non-proliferative DR (NPDR) is the early stage of DR, which is characterized by the presence of microaneurysms. Proliferative DR (PDR) is the more advanced stage of DR and can result in severe vision loss. Regular DR screening is important so that timely treatment can be implemented to prevent vision loss[2]. Early-stage intervention via glycemia and blood pressure control can slow down the progression of DR and late-stage interventions through photocoagulation or intravitreal injection can reduce vision loss[3]. In the United Kingdom and Iceland, where systematic national DR screening has been carried out, DR is no longer the leading cause of blindness among working-age adults[4,5]. Although routine DR screening is recommended by all professional societies, comprehensive DR screening is not widely performed[6–10], facing the challenges related to the availability of human assessors[3,11].

China currently has the largest number of patients with diabetes worldwide[12]. In 2016, the State Council issued the "Healthy China 2030" planning outline, which provided further guidance on the future direction of Chinese health reform[13]. The "Healthy China 2030" outlined the goal that all patients with diabetes will receive disease management and intervention by 2030. In China, there are about 40,000 ophthalmologists, with a 1:3000 ratio to patients with diabetes. As a cost-effective preventive measure, regular retinal screening is encouraged at the community level. Task shifting is one way the public health community can address this issue head-on so that ophthalmologists can do the treatment but not the screening. Task shifting is the name given by WHO to a process of delegation whereby tasks are moved, where appropriate, to less specialized health workers[14]. Recent evidence has established a role for screening by healthcare workers, given prior training in grading DR[3]. However, we still face the issues of insufficiency of their training and where they are placed in the system. Thus, diagnostic system using deep learning algorithms is required to help DR screening.

Recently, deep learning algorithms have enabled computers to learn from large datasets in a way that exceeds human capabilities in many areas[15–18]. Several deep learning algorithms with high specificity and sensitivity have been developed for the classification or detection of certain disease conditions based on medical images, including retinal images[19–23]. Current deep learning systems for DR screening have been predominantly focused on the identification of patients with referable DR (moderate NPDR or worse) or vision-threatening DR, which means the patients should be referred to ophthalmologists for treatment or closer follow-up[21,22,24]. However, the importance of identifying early-stage DR should not be neglected. Evidence suggests that proper intervention at an early stage to achieve optimal control of glucose, blood pressure, and lipid profiles could significantly delay the progression of DR and even reverse mild NPDR to DR-free stage[25].

In addition, the integration of these deep learning advances into DR screening is not straightforward because of some challenges. First, there are a few end-to-end and multi-task learning methods that can share the multi-scale features extracted from convolutional layers for correlated tasks, and further improve the performance of DR grading based on the lesion detection and segmentation, due to the fact that DR grading inherently relies on the global presence and distribution of the DR lesions[21,22,26–28]. Second, despite being helpful in DR screening, there are a few deep learning methods providing on-site image quality assessment with latency compatible with real-time use, which is one of the most needed additions at primary DR screening level and will have the impact on screening delivery at the community level.

Here we describe the development and validation of a deep learning-based DR screening system called DeepDR (Deep-learning Diabetic Retinopathy), which was a transfer learning assisted multi-task network to evaluate retinal image quality, retinal lesions, and DR grades. The system was developed using a real-world DR screening dataset consisting of 666,383 fundus images from 173,346 patients. In addition, we annotated retinal lesions, including microaneurysms, cotton-wool spots (CWS), hard exudates, and hemorrhages on 14,901 images, and used transfer learning[29] to enhance the lesion-aware DR grading performance. The system achieved high sensitivity and accuracy in the whole-process detection of DR from early to late stages.

## Results

**Data sources and network design**. DeepDR was developed using the fundus images of patients with diabetes who participated in the Shanghai Integrated Diabetes Prevention and Care System (Shanghai Integration Model, SIM) between 2014 and 2017 (Supplementary Table 1). A total of 666,383 fundus images from 173,346 patients with diabetes with integrity fundus examination records were enrolled in this study. Two retinal photographs (macular and optic disc centered)[30] were taken for each eye according to the DR screening guidelines of the World Health Organization[31]. Image quality (overall gradability, artifacts, clarity, and field), DR grades (non-DR, mild NPDR, moderate NPDR, severe NPDR, or PDR), and diabetic macular edema (DME) were labeled for each image. In addition, 14,901 images were labeled with retinal lesions, including microaneurysms, CWS, hard exudates, and hemorrhages.

Among the 173,346 subjects in the SIM cohort (referred as the local dataset in this study), 121,342 subjects (70%) were randomly selected as the training set, and the remaining 52,004 subjects (30%) served as the local validation set (Fig. 1). In the SIM cohort, each subject was enrolled only once and was recorded with the unique resident ID. So, the data separation was guaranteed between the training and local validation datasets. The prevalence of DR in the study cohorts is shown in Table 1. In the training dataset, 12.85% of images had DR, among which 27.94% were mild NPDR. In the local validation dataset of 200,136 images, 12.99% of images had DR, among which 27.30% were mild NPDR.

The DeepDR system consisted of three deep-learning sub-networks: image quality assessment sub-network, lesion-aware sub-network, and DR grading sub-network (Fig. 2). All the 466,247 images in the training dataset were used to train the image quality assessment sub-network to make binary classification of whether the image was gradable and recognize certain quality issues in terms of artifacts, clarity, and field problems of the retinal images; 415,139 images without quality issues were used to train the DR grading sub-network to classify the images into non-DR, mild NPDR, moderate NPDR, severe NPDR, or PDR, and binary classification of whether there was DME. The lesion-aware sub-network was trained using 10,280 images labeled with retinal lesions to achieve detection and segmentation of microaneurysms, CWS, hard exudates, and hemorrhages.

As shown in Fig. 2, our DeepDR system was designed as the transfer learning assisted multi-task network. Specifically, a DR base network was first pre-trained on ImageNet classification and then fine-tuned on our DR grading task using 415,139 retinal images. Next, we utilized transfer learning[32] to transfer the DR base network to the three sub-networks of the DeepDR system, rather than directly training randomly initialized sub-networks. During the process of transfer learning, we fixed the pre-trained weights in the lower layers of the DR base network and retrained the weights of its upper layers using

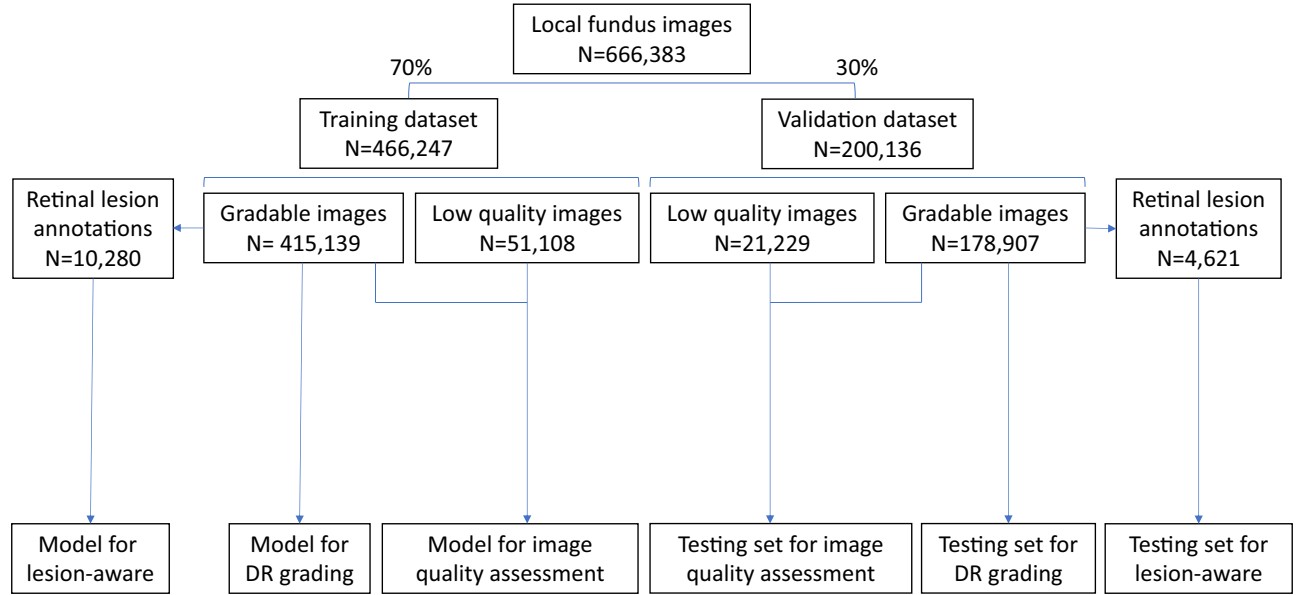

**Fig. 1 Data split in the local dataset (SIM cohort) for the training and local validation of the three sub-networks of the DeepDR system.** The local dataset was randomly divided into training or validation datasets. All 466,247 images in the training dataset were used for training the image quality assessment sub-network. The lesion detection sub-network was trained using 10,280 gradable images with retinal lesion annotations. Then, 415,139 gradable images in the training set were used for the training of the DR grading sub-network. All 200,136 images in the local validation dataset were used to test the image quality sub-network, and 178,907 gradable images were used to test the DR grading sub-network. Finally, 4621 gradable images labeled with retinal lesions were used to test the lesion detection sub-network. DR, diabetic retinopathy.

backpropagation. This process worked well since the features were suitable to all the DR-related learning tasks (evaluating image quality, lesion analysis, and DR grading). Furthermore, we concatenated the lesion features extracted by the segmentation module of the lesion-aware sub-network with the features extracted by the DR grading sub-network to enhance grading performance. To prevent the network from overfitting, an early stopping criterion[33] was used to determine the optimized number of iterations. For every task, we randomly split the training dataset into two parts, 80% of the data were used to train the network and the rest were used for early stopping. The network was tested on early stopping dataset every epoch during training and the performance of the network was recorded. If the area under the receiver operating characteristic curve (AUC) or intersect over union (IoU) increment was less than 0.001 for 5 epochs continuously, we stopped training and selected the best model as the final model.

**Performance of the DeepDR system.** The image quality assessment sub-network for assessing overall image quality and identifying artifacts, clarity, and field definition problems was tested using 200,136 images in the local validation dataset. DeepDR achieved an AUC of 0.934 (0.929–0.938) for overall image quality. For the identification of artifacts, clarity, and field definition issues, the system achieved AUCs of 0.938 (0.932–0.943), 0.920 (0.914–0.926), and 0.968 (0.962–0.973), respectively.

The lesion-aware sub-network was evaluated using 4621 gradable images with retinal lesion annotations from the local validation dataset. The results are shown in Fig. 3 and Supplementary Table 2. For microaneurysm, the AUC, sensitivity, specificity, and $F$-score were 0.901 (0.894–0.906), 88.0% (87.2–88.9%), 73.3% (72.0–74.3%), and 0.815, respectively. For CWS, the AUC, sensitivity, specificity, and IoU were 0.941 (0.935–0.946), 90.0% (87.9–91.9%), 83.1% (82.2–83.9%), and 0.711, respectively. For hard exudate, the AUC, sensitivity, specificity,

and IoU were 0.954 (0.949–0.957), 90.5% (88.9–91.5%), 85.8% (85.2–86.6%), and 0.971, respectively. For hemorrhage, the AUC, sensitivity, specificity, and IoU were 0.967 (0.965–0.969), 93.2% (92.6–94.1%), 88.0% (87.6–88.7%), and 0.738, respectively. The lesion-aware sub-network highlighted the lesion areas by masking the fundus images (Fig. 3B). To facilitate usability in clinical settings, a clinical report could be automatically generated for each patient (example report shown in Supplementary Fig. 1). This report showed the original fundus images with highlighted lesions, described the type and location of the retinal lesions along with DR gradings. In addition, we conducted an experiment to evaluate the utility of lesion-aware sub-network by measuring its effect on the grading accuracy of trained primary healthcare workers from community health service centers. Detailed study design is described in the Supplementary Information (Section "Supplementary Methods"). The results were tested using one-sided, two-sample Wilcoxon signed rank test and are shown in Table 2. The sensitivities of all DR grades and the specificity of severe DR were significantly improved with the aid of the DeepDR system. This suggested that visual hints of retinal lesions significantly improved the diagnostic accuracy of the primary healthcare workers, which can facilitate the task shifting of DR screening.

The DeepDR system achieved the whole-process diagnosis of DR from early to late stages based on the accurate detection of retinal lesions that was especially accurate for microaneurysms. In the local validation dataset, 178,907 gradable images were used to test the DR grading sub-network and the results are shown in Table 3. For the two images per eye, our DR grading sub-network made separate prediction per image, and then we accepted the more severe DR grade obtained from those images as the grading result for that eye, which was used to calculate the AUC of DR grades. The average AUC was 0.955 for DR grading. In particular, for mild NPDR, the AUC, sensitivity, and specificity were 0.943 (0.940–0.946), 88.8% (87.7–89.7%), and 83.9% (83.7–84.1%), respectively. For DME, the AUC was 0.946 (0.945–0.947), sensitivity was 92.8% (92.4–93.1%), and specificity was 81.3% (81.0–81.6%).

**Table 1 Summary of the development and validation datasets for diabetic retinopathy.**

| Source of datasets | Number | | | Number of images | | | | | |
|---|---|---|---|---|---|---|---|---|---|
| | Patients | Eyes | Images | Non-DR | Mild NPDR | Moderate NPDR | Severe NPDR | PDR | DME |
| Local dataset (SIM) | | | | | | | | | |
| Training dataset | 121,342 | 242,684 | 466,247 | 409,027 | 16,744 | 34,234 | 5007 | 1235 | 2711 |
| Validation dataset | 52,004 | 104,008 | 200,136 | 175,345 | 7100 | 15,065 | 2101 | 525 | 1215 |
| External validation datasets | | | | | | | | | |
| CNDCS | 23,186 | 46,372 | 92,672 | 77,619 | 3070 | 9681 | 1906 | 396 | – |
| NDSP | 6987 | 13,974 | 27,948 | 27,067 | 232 | 450 | 152 | 47 | – |
| EyePACS | – | – | 88,702 | 65,343 | 6205 | 13,153 | 2087 | 1914 | – |

*DR diabetic retinopathy, CNDCS China National Diabetic Complications Study, DME diabetic macular edema, NDSP Nicheng Diabetes Screening Project, NPDR non-proliferative diabetic retinopathy, PDR proliferative diabetic retinopathy, SIM Shanghai Integrated Diabetes Prevention and Care System (Shanghai Integration Model).*

**External validation**. To test the generalization of the system, we further evaluated the performance of DeepDR using two independent real-world cohorts and the publicly accessible dataset EyePACS for external validation. The first cohort was the China National Diabetic Complications Study (CNDCS) cohort, comprising 92,672 fundus images from 23,186 patients with diabetes and was acquired in 2018. The second cohort was the Nicheng Diabetes Screening Project (NDSP) cohort, comprising 27,948 fundus images from 6987 elderly subjects over 65 years of age and was acquired in 2018. The prevalence of diabetes was 31.7% in the NDSP cohort. The EyePACS dataset is a publicly available dataset from the United States, and consists of 88,702 fundus images.

The results for DR grading are shown in Table 3. In the CNDCS, the DeepDR system achieved AUCs of 0.916 (0.912–0.920) for mild NPDR, 0.927 (0.925–0.929) for moderate NPDR, 0.962 (0.959–0.965) for severe NPDR, and 0.955 (0.949–0.961) for PDR. In the NDSP and EyePACS dataset, the average AUCs for DR grading were 0.944 and 0.943, respectively. The system had high AUCs for mild NPDR, achieving 0.929 (0.916–0.942) and 0.937 (0.935–0.939) for the NDSP and EyePACS datasets, respectively.

**Real-time image quality feedback**. We employed DeepDR to provide real-time image quality feedback during the non-mydriatic retinal photography of 1294 elderly subjects from the NDSP cohorts (age over 65 years). Two retinal photographs (macular and optic disc centered) were taken of each eye. If DeepDR determined the quality of the first image of a field to be ungradable, a second image of the same field was recaptured. Only one more photograph was taken of each field to avoid contracted pupils due to the camera flash.

The results are shown in Table 4. During this process, 5176 retinal images were initially taken from 1294 patients. Of these, 1487 images (28.7%) were recognized as low-quality with artifacts, clarity, and/or field definition issues. Based on the feedback information, a second photograph was taken of these patients. For the 1487 initial low-quality images, 1065 (71.6%) recaptured images were of adequate quality. After replacing the low-quality images with recaptured images, the diagnostic accuracy of each grade of DR was improved. Especially for mild NPDR, the AUC increased from 0.880 (0.859–0.895) to 0.933 (0.918–0.950) ($P < 0.001$) and sensitivity increased from 78.5% (72.7–83.4%) to 87.6% (83.2–92.3%).

## Discussion

The DeepDR system achieved high sensitivity and specificity in DR grading. Rather than just generating a DR grading, it offers visual hints that help users to identify the presence and location of different lesion types. Introducing the image quality sub-network and lesion-aware sub-network into DeepDR improved the diagnostic performance and more closely followed the thought process of ophthalmologists. DeepDR can run on a standard personal computer with average-performance processors. Thus, it has great potential to improve the accessibility and efficiency of DR screening.

Several previous studies using deep learning approaches have been conducted on the detection of referable or vision-threatening DR detection. Gulshan et al. tested their deep learning system using 9963 fundus images and achieved a high level of performance for referable DR (AUC = 0.99)[21]. Ting et al. evaluated their deep learning system using 71,896 images and reported excellent results for referable and vision-threatening DR (AUCs of 0.936 and 0.958, respectively)[22]. Li et al. validated their deep learning system in a real-world multiethnic dataset of 35,201 images and achieved an AUC of 0.955 for vision-threatening

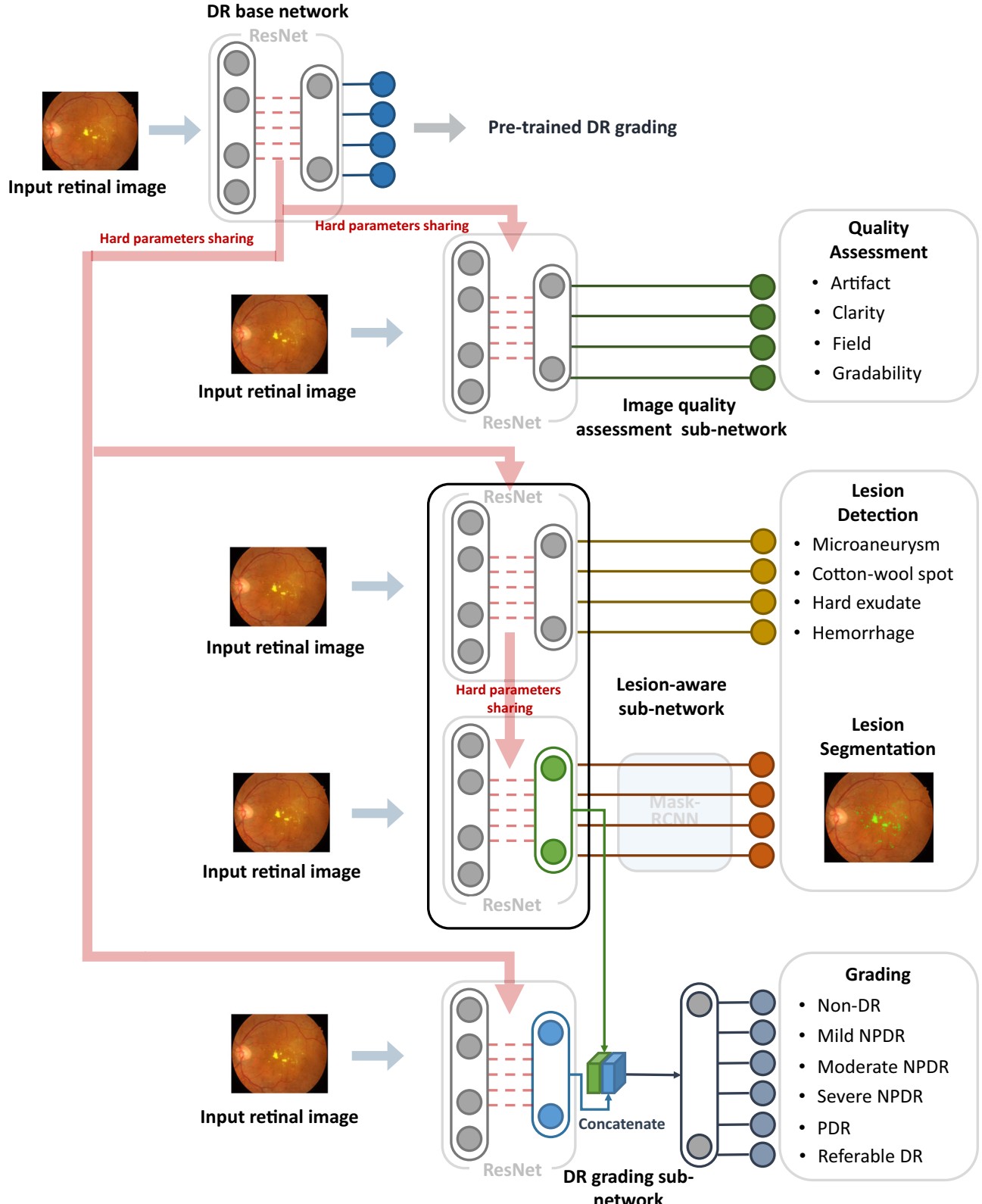

**Fig. 2 Visual diagram of the DeepDR system.** DeepDR system consisted of three sub-networks: image quality assessment sub-network, lesion-aware sub-network, and DR grading sub-network. We first pre-trained the ResNet to form the DR base network (top row). The trained weights of the pre-trained DR base network were then shared in the three different sub-networks of the system, indicated by the red arrow. These three sub-networks took retinal images as input and performed different tasks one-by-one. Furthermore, the lesion features extracted by the segmentation module of the lesion-aware sub-network (indicated by the green arrow) were concatenated with the features extracted by the DR grading sub-network (indicated by the blue arrow). DR, diabetic retinopathy; NPDR, non-proliferative diabetic retinopathy; PDR, proliferative diabetic retinopathy.

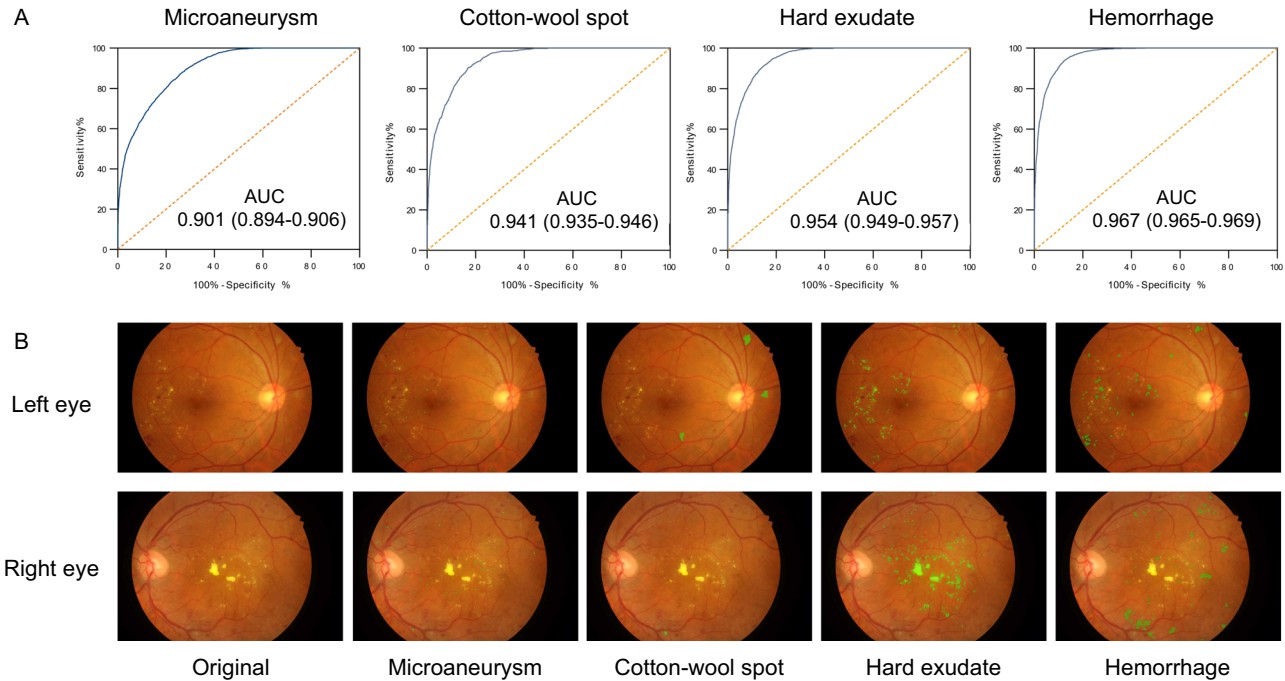

**Fig. 3 Performance of the lesion-aware sub-network. A** Receiver operating characteristic curve demonstrating the performance of the lesion-aware sub-network for retinal lesion detection (*n* = 4621). **B** Example images of retinal lesion segmentation: microaneurysms, cotton-wool spots, hard exudates, and hemorrhages are highlighted using green regions.

**Table 2 The sensitivity and specificity for DR grading by primary healthcare workers with or without the help of lesion detection and segmentation.**

| DR levels | Sensitivity | | | Specificity | | |
|---|---|---|---|---|---|---|
| | Unaided | Aided | *P* | Unaided | Aided | *P* |
| Non-DR | 0.586 (0.560–0.612) | 0.683 (0.679–0.686) | <0.001 | 0.924 (0.897–0.950) | 0.948 (0.937–0.959) | 0.504 |
| Mild NPDR | 0.589 (0.565–0.613) | 0.684 (0.680–0.688) | 0.001 | 0.893 (0.868–0.918) | 0.907 (0.896–0.918) | 0.362 |
| Moderate NPDR | 0.583 (0.554–0.613) | 0.680 (0.674–0.687) | 0.001 | 0.914 (0.893–0.936) | 0.922 (0.908–0.936) | 0.589 |
| Severe NPDR | 0.827 (0.822–0.833) | 0.843 (0.842–0.844) | 0.004 | 0.873 (0.852–0.894) | 0.935 (0.924–0.945) | 0.004 |
| PDR | 0.827 (0.823–0.831) | 0.843 (0.842–0.844) | <0.001 | 0.903 (0.882–0.924) | 0.922 (0.908–0.935) | 0.191 |

The sensitivity and specificity were tested using one-sided, two-sample Wilcoxon signed rank test.
*DR* diabetic retinopathy, *NPDR* non-proliferative diabetic retinopathy, *PDR* proliferative diabetic retinopathy.

DR[24]. Although these studies achieved excellent accuracy, they focused only on patients with referable DR who are then referred for specialist eye care. However, mild DR was classified into non-referable DR and was not distinguished from DR-free subjects[21,22,24].

The value of detecting early DR is underestimated, as there is little evidence that ophthalmic treatments, such as photocoagulation or anti-VEGF medications, are indicated at this stage[2]. Furthermore, if all the cases of DR are referred to ophthalmologists, it would likely overwhelm our medical systems. However, from the perspective of diabetes management, the screening for mild DR is of great clinical importance and may improve patients' outcomes. First, the identification of patients with mild DR facilitates health providers, such as family physicians, general practitioners, and endocrinologists, to participate in the patient education and management of blood glucose, lipid profiles, blood pressure, and other risk factors[2]. Secondly, there is no known cure for advanced DR, and some of the damage caused by leakage, oxygen deprivation, and blood vessel growth is permanent[34]. But there is evidence showing that optimal glycemic and blood pressure controls are strongly correlated with the

regression from mild DR to DR-free state[25], and intensive glycemic and lipid control reduces the rate of progression to vision-threatening DR[35]. Thirdly, screening for mild DR provides valuable information for clinical decision making. Although intensive glycemic control reduces the rate of photocoagulation, it increases the risk of severe hypoglycemia and incurs additional burden by way of polypharmacy, side effects, and cost[36]. The optimal glycemic target is controversial. The American College of Physicians guideline[37] set HbA1c levels 7–8% as the optimal target for most patients with diabetes, while the American Diabetes Association guideline[38] set the HbA1c target at 6.5–7.0%. Patients with mild DR could benefit from strict glycemic control[39]. Thus, the detection of mild DR can promote personalized diabetes management.

Accurate detection of microaneurysms is still a problem for deep learning systems[40]. In this study, to improve the performance of detecting specific retinal lesions and DR grading, we introduced an efficient retinal lesion-aware sub-network based on ResNet that avoided the problem of vanishing gradients, which made it a more sensitive feature extractor for small lesions compared to other existing network architectures (e.g., VGG and

**Table 3 Performance of the DeepDR system for diabetic retinopathy grading.**

| | | AUC (95% CI) | Sensitivity (95% CI) | Specificity (95% CI) | Positive likelihood ratio | Negative likelihood ratio |
|---|---|---|---|---|---|---|
| **Local validation** | | | | | | |
| SIM | Non-DR | 0.945 (0.943–0.947) | 0.912 (0.911–0.914) | 0.825 (0.818–0.831) | 5.201 (5.019–5.414) | 0.106 (0.104–0.109) |
| | Mild NPDR | 0.943 (0.940–0.946) | 0.888 (0.877–0.897) | 0.839 (0.837–0.841) | 5.511 (5.409–5.614) | 0.134 (0.122–0.146) |
| | Moderate NPDR | 0.955 (0.953–0.957) | 0.937 (0.931–0.943) | 0.837 (0.834–0.839) | 5.736 (5.647–5.824) | 0.075 (0.069–0.082) |
| | Severe NPDR | 0.960 (0.956–0.964) | 0.920 (0.904–0.936) | 0.860 (0.858–0.862) | 6.578 (6.418–6.732) | 0.093 (0.074–0.112) |
| | PDR | 0.972 (0.966–0.977) | 0.930 (0.898–0.958) | 0.895 (0.893–0.897) | 8.868 (8.526–9.190) | 0.078 (0.046–0.113) |
| | DME | 0.946 (0.945–0.947) | 0.928 (0.924–0.931) | 0.813 (0.810–0.816) | 4.955 (4.878–5.037) | 0.089 (0.084–0.094) |
| | Referable DR | 0.973 (0.972–0.974) | 0.941 (0.936–0.946) | 0.897 (0.895–0.899) | 9.157 (8.980–9.335) | 0.066 (0.061–0.071) |
| **External validation** | | | | | | |
| CNDCS | Non-DR | 0.937 (0.935–0.939) | 0.876 (0.874–0.878) | 0.831 (0.825–0.837) | 5.191 (5.019–5.382) | 0.149 (0.146–0.152) |
| | Mild NPDR | 0.916 (0.912–0.920) | 0.914 (0.904–0.923) | 0.743 (0.741–0.746) | 3.562 (3.507–3.615) | 0.116 (0.103–0.129) |
| | Moderate NPDR | 0.927 (0.925–0.929) | 0.890 (0.884–0.896) | 0.793 (0.790–0.796) | 4.304 (4.239–4.370) | 0.138 (0.131–0.146) |
| | Severe NPDR | 0.962 (0.959–0.965) | 0.918 (0.905–0.929) | 0.880 (0.878–0.882) | 7.624 (7.460–7.789) | 0.094 (0.080–0.108) |
| | PDR | 0.955 (0.949–0.961) | 0.927 (0.900–0.950) | 0.855 (0.852–0.857) | 6.377 (6.168–6.573) | 0.086 (0.059–0.117) |
| | Referable DR | 0.970 (0.969–0.971) | 0.940 (0.936–0.944) | 0.883 (0.881–0.885) | 8.050 (7.896–8.208) | 0.068 (0.063–0.073) |
| NDSP | Non-DR | 0.925 (0.915–0.934) | 0.907 (0.903–0.910) | 0.758 (0.729–0.787) | 3.750 (3.345–4.255) | 0.123 (0.117–0.129) |
| | Mild NPDR | 0.929 (0.916–0.942) | 0.905 (0.866–0.941) | 0.770 (0.765–0.775) | 3.942 (3.752–4.121) | 0.123 (0.076–0.175) |
| | Moderate NPDR | 0.957 (0.951–0.963) | 0.933 (0.910–0.956) | 0.861 (0.857–0.865) | 6.692 (6.443–6.940) | 0.077 (0.051–0.104) |
| | Severe NPDR | 0.953 (0.942–0.964) | 0.875 (0.820–0.927) | 0.885 (0.881–0.888) | 7.591 (7.066–8.103) | 0.141 (0.083–0.204) |
| | PDR | 0.956 (0.933–0.974) | 0.894 (0.795–0.976) | 0.900 (0.896–0.903) | 8.898 (7.920–9.753) | 0.118 (0.027–0.227) |
| | Referable DR | 0.965 (0.960–0.969) | 0.945 (0.926–0.962) | 0.860 (0.856–0.865) | 6.768 (6.528–7.004) | 0.064 (0.045–0.086) |
| EyePACS | Non-DR | 0.934 (0.932–0.936) | 0.900 (0.898–0.903) | 0.802 (0.797–0.807) | 4.546 (4.435–4.664) | 0.124 (0.121–0.127) |
| | Mild NPDR | 0.937 (0.935–0.939) | 0.903 (0.896–0.911) | 0.804 (0.801–0.807) | 4.613 (4.539–4.688) | 0.120 (0.111–0.129) |
| | Moderate NPDR | 0.932 (0.930–0.934) | 0.897 (0.892–0.903) | 0.802 (0.799–0.805) | 4.532 (4.459–4.606) | 0.128 (0.121–0.134) |
| | Severe NPDR | 0.954 (0.951–0.957) | 0.924 (0.913–0.935) | 0.848 (0.846–0.851) | 6.094 (5.977–6.214) | 0.089 (0.076–0.102) |
| | PDR | 0.961 (0.958–0.964) | 0.932 (0.920–0.943) | 0.862 (0.859–0.864) | 6.742 (6.607–6.884) | 0.079 (0.066–0.093) |
| | Referable DR | 0.946 (0.945–0.947) | 0.928 (0.924–0.931) | 0.813 (0.810–0.816) | 4.955 (4.878–5.037) | 0.089 (0.084–0.094) |

For each level, AUC is report in a one-vs-all manner, i.e., Mild NPDR means Mild NPDR vs. Non-DR, Moderate NPDR, Severe NPDR, and PDR. *DR* diabetic retinopathy, *NPDR* non-proliferative diabetic retinopathy, *PDR* proliferative diabetic retinopathy, *CNDCS* China National Diabetic Complications Study, *DME* diabetic macular edema, *NDSP* Nicheng Diabetes Screening Project, *SIM* Shanghai Integrated Diabetes Prevention and Care System (Shanghai Integration Model).

Inception)[41]. The lesion-aware sub-network contained feature pyramid structure that was designed to capture multi-scale features and mine the relationship of lesion types and position[42]. Meanwhile, transfer learning was used in our study and the lesion-aware sub-network contained the repurposed DR base network layers that were pre-trained by a base DR grading dataset of 415,139 retinal images. This boosted the performance of learning lesion detection and segmentation through the transfer of knowledge from DR grading task that has already been learned. As a result, the DeepDR system achieved AUCs of 0.901–0.967 for lesion detection, including microaneurysms, CWS, hard exudates, and hemorrhages. Retinal lesion detection and segmentation is of great clinical impact. Detecting different types of retinal lesions can provide guidance for clinical decision making. For example, fenofibrate may benefit patient with hard excaudate[43] and antiplatelet drugs should be used carefully in patient with retinal bleeding[44]. More importantly, one of the major problems in DR screening is detecting change or progression, as progression of retinal lesions is indicative of developing sight-threatening DR/DME[45–47]. Due to the fact that DR progression could be detected not only between different DR grades, but even within the same grade, our lesion-aware sub-network has the potential to capture tiny progression of certain kind of retinal lesions through follow-up of DR patients. Further studies are needed to evaluate this application in real-world clinical settings.

In previous studies, the deep learning systems were usually trained directly end-to-end from original fundus images to the labels of DR grades[21,22,24], these end-to-end systems might fail to encode the lesion features due to the black-box nature of deep learning[48]. In our study, instead of direct end-to-end training from fundus images to DR grades, an efficient lesion-aware sub-network was introduced to increase the ability of capturing lesion features. Due to the fact that embedding prior knowledge into the end-to-end machine learning algorithms can regulate machine learning models and shrink the search space[49], and the ophthalmologists read fundus images based on the presence of lesions, our DR grading network can leverage lesion features as prior knowledge to enhance the performance of DR grading. Previous studies, such as Michael D. Abràmoff et al.'s work[50], used multiple CNNs to detect hemorrhages, exudates, and other lesions, and those detected lesion results were used to classify referable DR by a classic feature fusion model. Differently, our DeepDR network was trained end-to-end with the features extracted from both the lesion-aware sub-network and the original image. In this way, our DR grading sub-network can further exploit the features to minimize the training error, thus improving grading results. As a result, DeepDR achieved a sensitivity of 88.8% and specificity of 83.9% for mild NPDR detection on the local validation dataset. Notably, DeepDR achieved the diagnosis of all stages of DR with sufficient accuracy in real-word datasets.

Despite the continuous optimization in digital fundus cameras, aging, experience, lighting, and other non-biological factors resulting from improper operation still results in high percentage of low-quality fundus images, and reacquisition is time-consuming and sometimes impossible[51,52]. Previous studies on image quality assessment have focused on post hoc image data processing[21,22]. In this study, a real-time image quality feedback sub-network was implemented to facilitate the DR screening. Based on the feedback information, the artificial intelligence-assisted image quality assessment can reduce the proportion of poor-quality images from 28.7% to 8.2%. Furthermore, with the improvement of image quality, the diagnostic accuracy was significantly improved, especially for mild DR. This real-time image quality feedback function allows the operators to identify image

**Table 4 Impact of real-time quality feedback on DR diagnosis using the DeepDR system.**

| | First image acquisition (low-quality images) n = 1487 | | | Second image acquisition n = 1487 | | | P* |
|---|---|---|---|---|---|---|---|
| | AUC (95% CI) | Sensitivity (95% CI) | Specificity (95% CI) | AUC (95% CI) | Sensitivity (95% CI) | Specificity (95% CI) | |
| Non-DR | 0.780 (0.761–0.805) | 0.633 (0.604–0.656) | 0.718 (0.682–0.750) | 0.912 (0.896–0.926) | 0.845 (0.815–0.868) | 0.798 (0.777–0.827) | <0.001 |
| Mild NPDR | 0.880 (0.859–0.895) | 0.785 (0.727–0.834) | 0.784 (0.769–0.804) | 0.933 (0.918–0.950) | 0.876 (0.832–0.923) | 0.785 (0.761–0.803) | <0.001 |
| Moderate NPDR | 0.921 (0.903–0.935) | 0.916 (0.874–0.947) | 0.759 (0.736–0.784) | 0.946 (0.932–0.957) | 0.791 (0.742–0.829) | 0.909 (0.886–0.926) | <0.001 |
| Severe NPDR | 0.936 (0.916–0.956) | 0.860 (0.779–0.932) | 0.832 (0.812–0.846) | 0.947 (0.928–0.959) | 0.753 (0.684–0.809) | 0.925 (0.910–0.938) | 0.070 |
| PDR | 0.934 (0.878–0.972) | 0.762 (0.625–0.947) | 0.872 (0.855–0.885) | 0.942 (0.908–0.970) | 0.905 (0.818–1.000) | 0.795 (0.775–0.824) | 0.401 |

*Two-sided P values were calculated for the comparison of AUC of first image collection vs. second image collection using binormal model methods. DR diabetic retinopathy, NPDR non-proliferative diabetic retinopathy, PDR proliferative diabetic retinopathy.

quality issue immediately and the patient does not need to be called back. It is a promising tool to reduce ungradable rate of the fundus images, thus increasing the efficiency of DR screening.

The limitation of this study is, firstly, the single-ethnic cohort used to develop the system. However, we used the publicly available EyePACS dataset from the United States for external validation and achieved satisfactory sensitivity and specificity. Secondly, the lesion-aware sub-network was tested only on the local validation dataset, because of the lack of lesion annotations in external cohorts. Further external validation in multiethnic and multicenter cohorts is needed to confirm the robustness of lesion detection and DR grading of the DeepDR system.

In conclusion, we developed an automated, interpretable, and validated system that performs real-time image quality feedback, retinal lesion detection, and early- to late-stage DR grading. With those functions, DeepDR system is able to improve image collection quality, provide clinical reference, and facilitate DR screening. Further studies are needed to evaluate deep learning system in detecting and predicting DR progression.

## Methods

**Ethical approval**. The study was approved by the Ethics Committee of Shanghai Sixth People's Hospital and conducted in accordance with the Declaration of Helsinki. Informed consent was obtained from participants. The study was registered on the Chinese Clinical Trials Registry (ChiCTR.org.cn) under the identifier ChiCTR2000031184.

**Image acquisition and reading process**. In the SIM project, retinal photographs were captured using desktop retinal cameras from Canon, Topcon, and ZEISS (Supplementary Table 1). All the fundus cameras were qualified by the organizer to ensure enough quality for DR grading. The operators of the cameras had all received standard training and the images were read by a centered reading group consisting of 133 certified ophthalmologists. The members in the reading group underwent training by fundus specialists and passed the tests. Original retinal images were uploaded to the online platform, and the images of each eye were assigned separately to two authorized ophthalmologists. They labeled the images using an online reading platform and gave the graded diagnosis of DR (Supplementary Fig. 2). The third ophthalmologist who served as the senior supervisor confirmed or corrected when the diagnostic results were contradictory. The final grading result was dependent on the consistency within these three ophthalmologists. At least 20% of the grading results would be randomly re-read to check the consistency. The total eligibility rate of spot-check was equal to or greater than 90%. If the reading group encountered difficult cases, they could apply for consultation from superior medical institutions. The overall disagreement rate in the SIM dataset was 18.9%. The primary cause of the diagnostic divergence was the decision between mild NPDR and non-DR.

For retinal lesion annotation, each fundus image was annotated by two ophthalmologists. For each type of lesion, two ophthalmologists generated two lesion annotations, respectively. We considered the two annotations to be valid if the IoU between them was greater than 0.85. Otherwise, a senior supervisor would check the annotations and give feedback to provide guidance. The image would be re-annotated by the two ophthalmologists until the IoU was larger than 0.85. Finally, we took the union of valid annotations as final ground truth segmentation annotation.

**Diagnostic criteria**. DR severity was graded into five levels (non-DR, mild NPDR, moderate NPDR, severe NPDR, or PDR, respectively), according to the International Clinical Diabetic Retinopathy Disease Severity Scale (AAO, October 2002)[53]. Mild NPDR was defined as the presence of microaneurysms only. Moderate NPDR was defined as more than just microaneurysms but less than severe NPDR, presenting CWS, hard exudates, and/or retinal hemorrhages. Severe NPDR was defined as any of the following: more than 20 intraretinal hemorrhages in each of the 4 quadrants; definite venous beading in 2+ quadrants; prominent intraretinal microvascular abnormalities (IRMA) in 1+ quadrant, and no signs of PDR. PDR was defined as one or more of the following: neovascularization, vitreous/preretinal hemorrhage[53]. DME was diagnosed if hard exudates were detected within 500 μm of the macular center according to the standard of the Early Treatment for Diabetic Retinopathy study[54]. Referable DR was defined as moderate NPDR or worse, DME, or both. Based on the guidelines for image acquisition and interpretation of diabetic retinopathy screening in China[55], the image quality was graded according to standards defined in terms of three quality factors, artifacts, clarity, and field definition[56], as listed in Table 5. The total score was equal to the score for clarity plus the score for field definition and minus the score for artifacts. A total score less than 12 was considered as ungradable.

**Table 5 Image quality scoring criteria.**

| Type | Image quality specification | Score |
|---|---|---|
| Artifact | No artifacts | 0 |
| | Artifacts are outside the aortic arch with scope less than 1/4 of the image | 1 |
| | Artifacts do not affect the macular area with range less than 1/4 | 4 |
| | Artifacts cover more than 1/4 but less than 1/2 of the image | 6 |
| | Artifacts cover more than 1/2 without fully covering the posterior pole | 8 |
| | Cover the entire posterior pole | 10 |
| Clarity | Only Level I vascular arch is visible | 1 |
| | Level II vascular arch and a small number of lesions are visible | 4 |
| | Level III vascular arch and some lesions are visible | 6 |
| | Level III vascular arch and most lesions are visible | 8 |
| | Level III vascular arch and all lesions are visible | 10 |
| Field definition | Do not include the optic disc and macula | 1 |
| | Only contain either optic disc or macula | 4 |
| | Contain optic disc and macula | 6 |
| | The optic disc or macula is outside the 1 papillary diameter and within the 2 papillary diameter range of the center | 8 |
| | The optic disc and macula are within 1 papillary diameter of the center | 10 |

Level I vascular arch was defined as the first bifurcations of major trunk veins; Level II vascular arch was defined as the veins deriving from the first bifurcation; Level III vascular arch was defined as the veins deriving from the second bifurcation. The total score was equal to the score for clarity plus the score for field definition and minus the score for artifact. A total score less than 12 was considered as ungradable.

**Architecture of the DeepDR system**. The DeepDR system had three sub-networks: image quality assessment sub-network, lesion-aware sub-network, and DR grading sub-network. Those sub-networks were developed based on ResNet[41] and Mask-RCNN[57]. Both ResNet and Mask-RCNN could be divided into two parts: (1) feature extractor, which took images as input and output features, (2) task-specific header, which took the features as input and generated task-specific outputs (i.e., classification or segmentation). Specifically, we chose to use the Mask-RCNN and ResNet with the same feature extractor architecture, so the feature extractor of one sub-network can be easily transferred to another.

The quality assessment sub-network can identify overall quality including gradability, artifacts, clarity, and field issues for the input images. To train the image quality assessment sub-network effectively, we initialized a ResNet with weights pre-trained on ImageNet and pre-trained the ResNet to form the DR base network. We utilized the weights of the convolution layers in the pre-trained DR base network to initialize the feature extractor of the image quality assessment sub-network. We assessed image quality in terms of multiple factors to determine if: (a) the artifact covered the macular area or the area of artifact was larger than a quadrant of the retinal image; (b) only Level II or wider vascular arch and obvious lesions could be identified (Level II vascular arch was defined as the veins deriving from the first bifurcation); (c) no optic disc or macula was contained in the image; and (d) the image was not gradable.

The lesion-aware sub-network can generate lesion presence and lesion segmentation masks of the input images. There were two modules in our lesion-aware sub-network: one was the lesion detection module and the other was the lesion segmentation module. The lesion detection module was a binary classifier that predicted whether any kind of lesions exist in a quadrant of the retinal image, as shown in Supplementary Fig. 3. The lesion segmentation module generated mask images to identify different lesions existing in the retinal images, as shown in Fig. 3B. We used ResNet and Mask-RCNN to form the lesion detection module and lesion segmentation module, respectively. Then we transferred the pre-trained DR base network to the lesion detection module by initializing the feature extractor of lesion detection module using the feature extractor of pre-trained DR base network, followed by fine-tuning the lesion detection module. Then we initialized the feature extractor of lesion segmentation module by reusing the feature extractor of the lesion detection module. The feature extractor layers of the lesion segmentation module were then fixed, and the rest of the layers of the module were updated during training. Non-maximum suppression was used in our lesion segmentation sub-module to select the bounding box with the highest objectiveness score from multiple predicted bounding boxes. Specifically, we first selected the bounding box with the highest objectiveness score, and then compared the IoU of this bounding box with other bounding boxes and removed the bounding boxes with IoU > 0.5. Finally, we moved to the next box with the highest objectiveness score and repeated until all boxes were either removed or selected.

The DR grading sub-network can fuse features from lesion-aware network and generate final DR grading results. To retain as much lesion information from the original retinal image as possible, we combined the pre-trained DR base network with the feature extractor of the lesion segmentation module in order to capture more detailed lesion features for DR grading. Then the weights in the extractors of DR grading sub-network were fixed, and the classification header of sub-network was updated during training.

The transfer learning assisted multi-task network was developed in our DeepDR architecture to improve the performance of DR grading based on lesion detection and segmentation. Due to the fact that DR grading inherently relies on the global presences of retinal lesions that contain multi-scale local texture and structures, the central feature of our multi-task learning method was designed to extract multi-scale features encoding local textures and structures of retinal lesions, where the transfer learning was used to improve the performance of DR grading task. Meanwhile, we used hard-parameter sharing in lesion-aware sub-network, and all the layers in the feature extractors of ResNet and Mask-RCNN are shared. Using hard-parameter sharing was important to reduce the risk of overfitting[58] due to the limited number of lesion segmentation labels. Besides, sharing the pre-trained weights can facilitate the training of both lesion detection task and lesion segmentation task. Additional experimental results demonstrated that hard-parameter sharing outperformed soft-parameter sharing for lesion segmentation is shown in Supplementary Table 3.

**Recommended computer configuration**. Any desktop or laptop computer with x86 compatible CPU, 10 GB or more of free disk space, and at least 8 GB main memory is capable to run the DeepDR system. There is no specialized hardware requirement, including GPU or any speed up card, to run the software. A powerful computer with more CPU cores and a GPU will speed up the diagnosis procedure significantly, while the diagnosis time on a typical laptop (i.e., with Intel I3 processor, no GPU, more than 8 GB memory) is also acceptable (less than 20 s per image).

**Statistical analyses**. The performances of DeepDR in assessing image quality, retinal lesion detection, and grading DR were measured by the AUC of the receiver operating characteristic curve generated by plotting sensitivity (the true-positive rate) versus 1-specificity (the false-negative rate). The operating thresholds for sensitivity and specificity were selected using the Youden index. The AUCs were compared using binormal model methods[59], where a two-sided $P$ value of less than 0.05 was considered statistically significant. For lesion detection, AUC was calculated as a binary classification to determine if a quadrant contained a certain kind of lesion. The performance of lesion segmentation was measured by IoU and $F$-score.

For CWS, hard exudates, and hemorrhages, we used IoU to measure the performance of segmentation network. The IoU was calculated as:

$$\text{IoU}(\mathbf{A}, \mathbf{B}) = \frac{|\mathbf{A} \cap \mathbf{B}|}{|\mathbf{A} \cup \mathbf{B}|} \tag{1}$$

where A and B were set of pixels in the retinal images (e.g., $\mathbf{A}$ was the segmented lesion and $\mathbf{B}$ was the ground truth).

For microaneurysms, the $F$-score was used instead of the IoU score, because the average diameter of microaneurysms in the retinal image was usually less than 30 pixels, minor change in the predicted map would result in a large change in IoU score. $F$-score was calculated as:

$$F = \frac{2 \bullet |\text{tp}|}{2 \bullet |\text{tp}| + |\text{fp}| + |\text{fn}|} \tag{2}$$

P was the set of all predicted microaneurysms produced by the network, G was the set of all microaneurysms annotated by ophthalmologists. $\text{tp} = (\mathbf{p} \in \mathbf{P}, |, \exists \mathbf{g} \in \mathbf{G}, \text{IoU}(\mathbf{p}, \mathbf{g}) \geq 0.5)$ represented the set true-positive predicts of microaneurysms, $\text{fp} = (\mathbf{p} \in \mathbf{P}, |, \forall \mathbf{g} \in \mathbf{G}, \text{IoU}(\mathbf{p}, \mathbf{g}) < 0.5)$ represented the set false-positive predicts of microaneurysms, $\text{fn} = (\mathbf{g} \in \mathbf{G}, |, \forall \mathbf{p} \in \mathbf{P}, \text{IoU}(\mathbf{p}, \mathbf{g}) < 0.5)$ represented the set of false-negative predictions of microaneurysms. $|\bullet|$ represented the cardinality (size) of a set.

Python version 3.7.1 (Python Software Foundation, Delaware, USA) was used for all statistical analyses in this study. The following third-party python packages were used: OpenCV version 2.4.3 (Intel Corporation, California, USA) was used for image loading and decoding image. Pytorch version 1.0.1 (Facebook, Massachusetts, USA) was used for convolutional neural network computing. Scikit-learn version 0.20.0 (David Cournapeau, California, USA) was used for calculating AUC. Pandas version 0.23.4 (Wes McKinney, Connecticut, USA) was used for loading ground truth and metadata. NumPy version 1.15.4 (Travis Oliphant, Texas, USA) was used for calculating IoU and $F$-score.

**Reporting summary**. Further information on research design is available in the Nature Research Reporting Summary linked to this article.

## Data availability

The export of human-related data is governed by the Ministry of Science and Technology of China (MOST) in accordance with the Regulations of the People's Republic of China on Administration of Human Genetic Resources (State Council No.717). Request for the non-profit use of the fundus images and related clinical information in the SIM, NDSP, and CNDCS cohorts should be sent to corresponding author Weiping Jia. The joint application of the corresponding author together with the requester for the data sharing will be generated and submitted to MOST. The data will be provided to the requester after the approval from MOST. The EyePACS dataset is publicly available at https://www.kaggle.com/c/diabetic-retinopathy-detection/data. The rest of the data are available from the corresponding author upon reasonable request.

## Code availability

The code being used in this study for developing the algorithm is provided at https://zenodo.org/badge/latestdoi/334570111.

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

## Acknowledgements

We would like to thank all the medical staffs and study subjects who participated in the Shanghai Integrated Diabetes Prevention and Care System Study. This work was supported by Shanghai Municipal Grants Award (GWIV-3), National Natural Science Foundation of China (NSFC) - National Health and Medical Research Council of Australia (NHMRC) joint research grant (81561128016), Shanghai Belt and Road Joint Laboratory of Intelligent Diagnosis and Treatment of Metabolic Diseases (18410750700), Science and Technology Innovation Action Plan from Shanghai Science and Technology Commission (17411952600) and Shanghai Municipal Key Clinical Specialty to W.J., and General Project from NSFC (61872241) to B.S., and Science and Technology Innovation Action Plan from Shanghai Science and Technology Commission (16DZ0501100) to Q.W.

## Author contributions

W.J., B.S., and H.Z. conceived and oversaw overall direction. L.D. designed the deep learning algorithm and the computational framework. L.W., H.L., and L.D. designed the study, interpreted the results and drafted the manuscript. C.C., Q.W., H.K., X.W., X.H., Y.L., L.L., and X.L. collected and organized data. Y.S., Y.W., and R.L contributed to data analysis. Q.W., Y.C., D.S., and X.Y. provided critical comments and reviewed the manuscript.

## Competing interests

The authors declare no competing interests.
