## [Peer Review File · Nature Communications]

REVIEWER COMMENTS

Reviewer #1 (Remarks to the Author):

Overall Paper Goal

The authors aim to develop and validate a deep learning model to detect early-to-late DR. Overall, the authors have done a great job explaining the methodology, results and limitations. The model shows promising results on the local population, and is generalizable to another patient population. The authors do a good job of explaining machine learning concepts for non-ML readers. However, the paper has some shortcomings in regards to clinical impact, algorithm development and manuscript clarity. We have summarized the following points which should be clarified or fixed.

Clinical Impact

The primary aim of the paper seems to be the importance of detecting mild DR. Adding more supporting evidence may strengthen this case. Additionally, the clinical impact of lesion segmentation is not explained or justified. Alternatively the paper could focus its primary theme around other meta areas e.g. model developed on a particular population is generalizable to other populations.

The clinical impact of detecting mild DR on preventing blindness is not clear. The authors claim that detecting mild DR will enable better self-management of diabetes. This needs to be supported with references, or the speculative nature of this comment needs to be highlighted in the paper.

The study focuses on only three lesions for lesion detection. Some explanations around why these lesions were selected over others would provide clarity. In lines 145-150, the list of lesions is incomplete: e.g. Heme, CWS, Exudates are not mentioned. Especially when the system has lesion detection focused on Heme, Exudates and MAs.

The manuscript mentions labeling pixel-level retina lesions. The authors could clarify how pixel-level annotations were achieved since humans cannot feasibly label individual pixels. Consider revising to describe as annotated lesions vs pixel-level annotation (line 108). Furthermore, in lines, 183-184: it is not clear if annotation or classification was used for lesions labeling as it concerns performance metrics. Were pixel-level lesions labelled by ophthalmologists, and how were disagreements on lesions resolved?

Data splits

The split of patients is based on data collection districts which does not guarantee data separation between train, tune, and test sets. Consider adding some details around how data separation was maintained as patients could move between district lines (line: 136-138). In addition, the split by district raises data uniformity concerns. Does division by district cause dataset shift, for example, different camera types, image quality, etc.? A superior split method may be splitting proportionately across all districts in aggregate. Another option to explore is using multi-fold cross-validation to ensure algorithm correctness across multiple splits.

Grading

At a high level, the sections describing grading require deeper consideration and much clarification.

The description for image gradability could be further clarified in lines 154-157: by explaining how the image gradability is classified if macula or optic disc is partially visible in an image. Consider specifying thresholds for extent of visibility required to classify an image as gradable.

The image grading process has some inconsistencies between the supplement and main paper. Consider revising the "Image reading process" in the supplement. The supplement mentions review by supervisor and lead reader suggesting two overreads but the manuscript mentions one senior grader for overreading when there is disagreement between the first two graders.

The number of graders is unclear. Table S1 in the supplement describes grader expertise. It is unclear if this describes the pool of graders or the number of graders per image. Were only two graders involved for the local dataset? If so, the training data is highly biased to these two graders and especially the senior retinal specialist. If not, then was the NSCD dataset graded by all 7 graders for every image?

Algorithm

More in-depth explanation can be provided to explain how the lesion detection model was developed and evaluated. For example, what similarity measure between test segmentation and ground truth are they using when calculating AUC (binary lesion presence? Overlap of ROIs?) It seems the lesion detection module was only tested on the "local" dataset. If this is true, it is important for the paper to acknowledge this limitation.

The authors describe multi-task learning as one of the differentiating factors from previous studies. However, it is not clear which hidden layers are shared, whether hard or soft parameters sharing is deployed and why. The authors mention the use of transfer learning from one model to another in several places throughout the paper; it is not clear whether tasks are learned one after another, or simultaneously. Consider adding a diagram to visualize the model architecture.

Manuscript Clarity

Line 90: the authors mentioned previous studies have “difficulties identifying early-stage DR.” Is there evidence that supports this statement? Or were previous studies focusing on referable DR rather than early-stage DR?

Line 127: Consider linking Table S1 list of cameras.

Line 145: Consider adding citation to ICDR.

Line: 180: Maybe reference Table 1 in this section.

Line 187-189: Consider more details for NSCD and NDSP. Were they non-diabetic?

Line: 267-269: Maybe quantify standard average performance computer.

Line: 319: Unclear what 30% correlates to.

Reviewer #3 (Remarks to the Author):

Thank you for submitting your paper to this journal.

This is an important topic to cover and one that requires large number of images both for training and for validation; these requirements are met in this report. I have a few questions and remarks for the authors.

1, Abstract: retinal screening does not prevent blindness, it identifies the patients who need to have treatment; the treatment will prevent blindness. Timely referral and treatment resulting from good quality screening means higher likelihood of preventing blindness and so pls rephrase this sentence.

2, Second sentence of the abstract is not accurate: we do have an accurate early detection system (human graders) and they are NOT incapable, that would be an insult to the whole workforce in retinal screening. Please remove or completely re-phrase this sentence. I suspect you are wanting to say that there are not enough appropriately trained healthcare workers to carry out the task of looking at images of all patients with diabetes on a timely manner? However, the way it is presented does not convey this message.

3, The abstract has no aim and very little results in it; "high accuracy" is not enough to let the reader know what the DeepDR system can achieve.

4, What is comparable accuracy to external validation set?

5, The last sentence of the abstract is not supported by data or previous evidence; DR screening itself does not produce better healthcare outcomes; acting upon results might.

6, Body of the paper: Diabetic retinopathy (DR) - please use the abbreviation right from the start of the text.

7, China has the largest population of diabetes in the world. - please re-write this sentence.

8, It would be important to acknowledge that in countries where systematic DR screening has been carried out, DR is not the leading cause of blindness amongst adults anymore (Iceland and UK), as this concept is important for your arguments later on.

9, "Few communities have qualified ophthalmologists to carry out DR screening. Patients with severe DR at diagnosis have missed the best treatment opportunity, which leads to poor prognosis or even blindness." - these are two different concepts. Most countries have far too few ophthalmologists and screening is mostly NOT carried out by ophthalmologist, the key issue is to task-shift/task-share so ophthalmologists can do the treatment and not the screening.

DR is NOT treated in early stages and in many cases you can save sight even for PDR patients as long as the sight is good at the start of the treatment, and so this statement is not correct.

10, You argue that MAs must be diagnosed so appropriate public health and general diabetes care mechanisms are put in place for the patient to delay the onset of visual loss by better control of

diabetes. This is an important concept, but you have to draw it out in the discussion that most of these procedures will not depend on the eyecare providers at this stage of the diabetic eye disease.

11, Methods: pls clarify what "grade 2 vascular arch" means, some readers will not be familiar with the terminology and understanding this concept is vital.

12, "because it was a community-based cohort rather than a diabetic patients-based cohort". This half a sentence does not belong here and really does not necessarily explain the situation. Either re-word or remove.

13, What difference did the age-difference between cohorts make? Was there a difference in duration of diabetes as well?

14, The real-time image feedback is a great feature with far-reaching consequences in the world of screening. If the screeners know immediately if the images are good quality or not, then they will be able to take a photo then and there and the patient does not need to be called back, saving time and effort for all parties. This must be discussed appropriately in the discussion as reducing ungradable rate is one of the key issues in DR screening programmes.

15, Line 250: "especially for microaneurysm". I am not sure that this is entirely appropriate to word this way. All other AI based programmes are good at picking up MAs. Many softwares that grade for "referrable DR" are indeed based on identification of MAs and many over-refer as the criteria is set low so they do not miss any patients who do require treatment. The same concept of problematic MA detection surfaces from line 293, this is not really raised by others beforehand in the AI field as problematic.

16, It would be important to provide details as to what drove the disagreements in terms of what levels of retinopathy/DME were missed and why. It can be just as informative as where there was agreement and it would give clarity to this aspect. How did you ensure that the human grading was good quality? What was the level of agreement there?

17, The one topic that is not discussed at all and is one of the major problems in DR screening is detecting change or progression. We know from the UK data that DR progression is indicative of developing sight threatening DR/DME within 4 years and so for an AI software to be able to detect change is imperative. I appreciate that this might be outside of the scope of this paper, but definitely would have to feature in the discussion and the way forward section.

Point-by-point responses (Manuscript No: NCOMMS-19-40862B)

We are thankful for the opportunity to review our revised manuscript. We thank all the reviewers for their insightful and constructive comments. We have now revised the manuscript to address the reviewers' concerns. Our point-by-point responses are following:

Responses to Reviewer #1

Overall Paper Goal

The authors aim to develop and validate a deep learning model to detect early-to-late DR. Overall, the authors have done a great job explaining the methodology, results and limitations. The model shows promising results on the local population, and is generalizable to another patient population. The authors do a good job of explaining machine learning concepts for non-ML readers. However, the paper has some shortcomings in regards to clinical impact, algorithm development and manuscript clarity. We have summarized the following points which should be clarified or fixed.

Clinical Impact

The primary aim of the paper seems to be the importance of detecting mild DR. Adding more supporting evidence may strengthen this case. Additionally, the clinical impact of lesion segmentation is not explained or justified. Alternatively the paper could focus its primary theme around other meta areas e.g. model developed on a particular population is generalizable to other populations.

Answer: We appreciate so much for this comment. The ability to detect mild DR is one of the advantages of our system. In this version, the clinical impact has been clarified and more evidences have been added to strengthen this case.

- a) The identification of patients with mild DR facilitates health providers, such as family physician, general practitioners and endocrinologists, to participate in the patient education and management of blood glucose, lipid profiles, blood pressure and other risk factors (*Diabetes Care, 2020,43(Supplement 1): S135-51*).
- b) There is no known cure for advanced DR, and some of the damage caused by leakage, oxygen deprivation, and blood vessel growth is permanent (*American Journal of Ophthalmology, 2005,139(2):290-4*). But there is evidence showing that optimal glycemic and blood pressure controls are strongly correlated with the regression from mild DR to DR-free state (*Diabetes Care, 2013,36(12):3979-84*), and intensive glycemic and lipid control reduces the rate of progression to vision-threatening DR (*New England Journal of Medicine, 2010,363(3):233-44*).

- c) Screening for mild DR provides valuable information for clinical decision making. Although intensive glycemia control reduces the rate of photocoagulation, it increases the risk of severe hypoglycemia and incurs additional burden by way of polypharmacy, side effects, and cost (*BMJ*, 2019,367:15887). The optimal glycemic target is controversial. The American College of Physicians guideline set HbA1c levels 7-8% as the optimal target for most diabetic patient (*Annals of internal medicine*, 2018, 168(8): 569-76), while the American Diabetes Association guideline set the HbA1c target at 6.5-7.0% (*Diabetes Care*, 2020,43(Suppl 1): S66). The patients with mild DR could benefit from restrict glycemic control. Thus, the detection of mild DR can promote the personalized diabetes management.

These above evidences have been added to the Discussion (Page 11, Line 217-230)

For lesion detection and segmentation, we summarized two major clinical impacts:

- a) One of the major problems in DR screening is detecting change or progression, as progression of retinal lesions is indicative of developing sight threatening DR/DME (*Diabetes care*, 2013, 36(5): 1254-59; *Acta Ophthalmologica Scandinavica*, 2004, 82(6):679-85; *Archives of Ophthalmology*, 2001, 119(4): 547-53). Due to the fact that DR progression could be detected not only between different DR grades, but even within the same grade, our lesion-aware sub-network has the potential to capture tiny progression of certain kind of retinal lesion through follow-up of DR patients. Further studies are needed to evaluate this application in real-world clinical settings.
- b) The detection of different types of retinal lesions can provide guidance for clinical decision making. For example, fenofibrate may benefit patient with hard excaudate (*The Lancet*, 2007,370,1687-97) and antiplatelet drugs should be used careful in patient with retinal bleeding (*Ophthalmology*, 2016,123(2):352-60).

These clinical impacts of lesion detection and segmentation have been added to the Discussion (Page 12, line 242-249).

The clinical impact of detecting mild DR on preventing blindness is not clear. The authors claim that detecting mild DR will enable better self-management of diabetes. This needs to be supported with references, or the speculative nature of this comment needs to be highlighted in the paper.

Answer: Thank you for the comment. This is what we observed from our clinical experience that patients who are aware of the condition of their complications tend to pay more attention on glucose monitoring, which may play a positive role in the self-management of diabetes. Based on your suggestion, this point needs more direct

data to support. We removed the related sentence from the revised manuscript. We will conduct further study regarding early diagnosis of DR and self-management of diabetes in the future.

The study focuses on only three lesions for lesion detection. Some explanations around why these lesions were selected over others would provide clarity. In lines 145-150, the list of lesions is incomplete: e.g. Heme, CWS, Exudates are not mentioned. Especially when the system has lesion detection focused on Heme, Exudates and MAs.

Answer: Thank you for the suggestion. We have completed the list of lesions in the diagnostic criteria. Mild NPDR was defined as the presence of microaneurysms only. Moderate NPDR was defined as more than just microaneurysms but less than severe NPDR, presenting cotton-wool spots (CWS), hard exudates, and/or retinal hemorrhages. Severe NPDR was defined as any of the following: more than 20 intraretinal hemorrhages in each of the 4 quadrants; definite venous beading in 2+ quadrants; prominent intraretinal microvascular abnormalities (IRMA) in 1+ quadrant, and no signs of PDR. PDR was defined as one or more of the following: neovascularization, vitreous/preretinal hemorrhage (*Ophthalmology*, 2003,110,1677-82) (Page 15, Line 310-315).

Regarding why we selected these lesions, first, microaneurysm, CWS, hard exudate, and hemorrhage are common lesions in DR. The amount of training data is large enough. Secondly, mild NPDR is characterized by microaneurysm, and CWS, hard exudate and hemorrhage are important for the grading of NPDR. These are the reasons we chose these four lesions for lesion detection. According to your suggestion, now the system is developed to detect four lesions: microaneurysm, CWS, hard exudate, and hemorrhage (Fig. 3 as below).

Fig. 3: Retinal lesion detection and segmentation.

(A) Performance of the lesion-aware sub-network for retinal lesion detection. (B) Example images of retinal lesion segmentation: microaneurysms, cotton-wool spots, hard exudates, and hemorrhages are highlighted using green regions.

The manuscript mentions labeling pixel-level retina lesions. The authors could clarify how pixel-level annotations were achieved since humans cannot feasibly label individual pixels. Consider revising to describe as annotated lesions vs. pixel-level annotation (line 108).

Answer: Thank you for the comments. The pixel-level labeling in our previous manuscript means that ophthalmologists can use an DR image annotation software tool that we developed (similar to LabelMe <http://labelme.csail.mit.edu>), to create DR lesion annotations (binary color images with white color regions indicating the lesion area and black color regions indicating background, as shown in Fig. R1-a and Fig. R1-b). By using this annotation tool, the ophthalmologists can use different shapes (polygon, rectangle, circle, line, and point) to easily mark the lesions in the DR image, as shown in Fig. R1-c the example of annotating hemorrhage. According to your suggestion, we have changed the description “pixel-level labeling” to “lesion annotation” throughout the paper for more accurate expression.

Fig. R1: Illustration of DR image annotation software tool.

a) Original fundus image, b) A sample of annotated lesions, white color indicates hemorrhage, black color indicates background, c) Illustration of the annotating the hemorrhage region.

Furthermore, in lines, 183-184: it is not clear if annotation or classification was used for lesions labeling as it concerns performance metrics. Were pixel-level lesions labelled by ophthalmologists, and how were disagreements on lesions resolved?

Answer: The DeepDR system is a multi-task learning system that has two machine learning tasks related to DR lesions, one is lesion detection, the other is lesion segmentation. The task of lesion detection is to classify whether there is certain kind of lesion existing in the specific region in the retina image (in our study, we defined the specific region as a quadrant of a retina image, as shown in Supplementary Fig. 3). For lesion detection task, the algorithm was trained using binary classification labels that indicate whether a certain type of lesion exists in the quadrant of the retina image. For this task, we used AUC to evaluate the performance of lesion detection. The task of lesion segmentation is to mark different types of lesions by different sets of pixels in the image space. The task of lesion segmentation was trained using the “lesion annotations” mentioned above. To evaluate the segmentation performance of hemorrhages, hard exudates, and cotton-wool spots, we used Intersection over Union (IoU) to quantify the percentage of overlap area between the lesion annotation and output (*Proceedings of the British Machine Vision Conference, 2013, pp. 32.1–32.11*). Meanwhile, we used F-score (*European Conference on Information Retrieval, 2005, 345–359*) to evaluate the segmentation performance of microaneurysms, where

F-score is defined as the harmonic mean of the precision and recall. We have provided more details to describe how AUC, IoU and F-score were calculated in the revised Method. (Page 18-19, Line 369-382)

Supplementary Fig. 3: Illustration of lesion detection task.

Lesion detection module was a binary classifier that predicted whether any kind of lesions exists in a quadrant (1/4 of the image) of the retinal image.

For retinal lesions annotation, each fundus image was annotated by two ophthalmologists. For each type of lesion, two ophthalmologists generated two lesion annotations, respectively. We consider the two annotations to be valid if the IoU between them is greater than 0.85. Otherwise, a senior supervisor would check the annotations and give feedback to provide guidance. The image would be re-annotated by the two ophthalmologists until the IoU was larger than 0.85. Finally, we take the union of valid annotations as final ground truth segmentation annotation. We have added those descriptions about lesion annotation and how the disagreements resolved have been added in the Method (Page 14, Line 300-305).

Data splits

The split of patients is based on data collection districts which does not guarantee data separation between train, tune, and test sets. Consider adding some details around how data separation was maintained as patients could move between district lines (line: 136-138). In addition, the split by district raises data uniformity concerns. Does division by district cause dataset shift, for example, different camera types, image quality, etc.? A superior split method may be splitting proportionately across all districts in aggregate. Another option to explore is using multi-fold cross-validation to ensure algorithm correctness across multiple splits.

Answer: Thank you for pointing out this important question. It is essential to guarantee no data overlap between training and testing dataset. In our study, we used the data from the Shanghai Integrated Diabetes Prevention and Care System (Shanghai Integration Model, SIM) as the local dataset. The project aimed to perform a city-wide screening for diabetic complications. Each patient was enrolled only once and was recorded with the unique resident ID. So, the data separation was guaranteed between the training and local testing datasets. We have added the details in the manuscript (Page 6, Line 115-117).

For the concern of the data uniformity, the SIM project was well-organized to reduce the possibility of data shift between districts. The operators had all received standard training and the images were read by a centered reading group consist of 133 certified ophthalmologists. We have added the detailed description in the Method (Page 14, Line 288-290).

According to the reviewer's suggestion, we conducted additional experiment by re-training the system using splitting proportionately across all districts in aggregate. We randomly selected 80% of all patients into training set and the rest 20% into testing set. The local test results are shown in Table R1. We also validate the robustness of results using multi-fold cross-validation (8:2), and results are shown in Table R2. Compared with DR grading results in our manuscript, the performance of DeepDR system trained with randomly split dataset or cross-validation are similar.

Table R1. Local and external test for the performance of DR grading using randomly split method.

Local validation		AUC (95% CI)	Sensitivity (%) (95% CI)	Specificity (%) (95% CI)
SIM*	Non-DR	0.945(0.941-0.949)	84.7(83.9-85.2)	88.9(88.0-89.7)
	Mild NPDR	0.943(0.938-0.947)	86.6(85.7-87.6)	85.7(85.0-86.4)
	Moderate NPDR	0.955(0.956-0.962)	89.9(88.9-90.9)	88.6(88.2-89.4)
	Severe NPDR	0.960(0.955-0.965)	91.5(89.7-93.2)	87.2(86.6-87.9)
	PDR	0.972(0.969-0.977)	92.6(90.4-95.6)	91.6(91.1-92.0)
	DME	0.928(0.919-0.934)	79.2(76.6-81.7)	89.4(88.9-90.0)
	Referable DR	0.973(0.969-0.977)	91.0(89.1-92.6)	91.1(90.6-91.6)
External validation				
CNDCS*	Non-DR	0.937(0.933-0.941)	86.0(85.3-86.7)	86.2(85.2-87.2)
	Mild NPDR	0.916(0.911-0.921)	84.2(83.2-85.2)	83.7(83.0-84.5)
	Moderate NPDR	0.927(0.922-0.931)	84.2(83.1-85.4)	84.6(83.9-85.3)
	Severe NPDR	0.962(0.956-0.967)	88.0(86.0-90.0)	88.4(87.8-88.9)
	PDR	0.955(0.947-0.962)	89.5(86.4-92.4)	90.4(89.9-90.9)
	Referable DR	0.970(0.967-0.973)	89.8(88.8-90.7)	90.3(89.7-90.9)
NDSP	Non-DR	0.925(0.913-0.936)	83.3(82.2-84.4)	84.5(81.2-87.6)
	Mild NPDR	0.929(0.916-0.941)	85.7(82.6-88.7)	84.9(83.8-85.9)
	Moderate NPDR	0.957(0.945-0.967)	86.8(82.8-90.4)	87.3(86.3-88.3)
	Severe NPDR	0.953(0.932-0.972)	85.1(78.1-91.6)	85.0(84.0-85.9)
	PDR	0.956(0.933-0.976)	87.5(78.1-95.7)	88.1(87.1-89.0)
	Referable DR	0.965(0.955-0.974)	88.5(85.0-91.9)	89.9(89.0-90.8)
EyePACS	Non-DR	0.934(0.930-0.938)	85.6(84.9-86.3)	84.8(83.8-85.8)
	Mild NPDR	0.937(0.933-0.941)	86.8(85.8-87.7)	83.8(83.1-84.6)
	Moderate NPDR	0.932(0.928-0.936)	85.6(84.5-86.8)	85.3(84.6-86.0)
	Severe NPDR	0.954(0.948-0.960)	86.3(84.2-88.3)	86.5(85.9-87.1)
	PDR	0.961(0.953-0.968)	89.8(86.7-92.7)	89.6(89.1-90.1)
	Referable DR	0.946(0.942-0.950)	86.7(85.6-87.8)	86.2(85.5-86.8)

* According to the expert consensus on naming notation from the Chinese Diabetes Society, we changed the notations of the two studies. Shanghai Integrated Diabetic Prevention and Care System Study (SIDPCSS), as referred in the previous version of manuscript, is now called Shanghai Integrated Diabetic Prevention and Care System (Shanghai Integrated Model, SIM). Nationwide Screening for Complications of Diabetes (NSCD), as referred in the previous version of manuscript, is now called China National Diabetic Complications Study (CNDCS).

Table R2. Local and external test for the performance of DR grading using multi-fold cross-validation.

Local validation		AUC (95% CI)	Sensitivity (%) (95% CI)	Specificity (%) (95% CI)
SIM*	Non-DR	0.943(0.939-0.947)	84.4(83.7-85.0)	89.7(88.9-90.5)
	Mild NPDR	0.944(0.940-0.949)	86.3(85.9-87.8)	85.5(84.6-86.0)
	Moderate NPDR	0.960(0.959-0.965)	89.3(88.4-90.4)	88.0(87.6-88.8)
	Severe NPDR	0.962(0.957-0.967)	91.3(89.8-93.3)	86.3(86.0-87.3)
	PDR	0.966(0.964-0.972)	92.3(90.2-95.4)	91.7(91.2-92.1)
	DME	0.924(0.915-0.930)	79.5(76.9-82.0)	89.7(89.2-90.3)
	Referable DR	0.967(0.964-0.972)	90.9(88.9-92.4)	91.4(90.8-91.8)
External validation				
CNDCS*	Non-DR	0.934(0.930-0.938)	85.7(85.3-86.2)	85.9(85.4-86.4)
	Mild NPDR	0.914(0.909-0.920)	84.0(83.8-84.3)	83.5(83.2-83.9)
	Moderate NPDR	0.926(0.922-0.929)	84.1(83.8-84.4)	84.5(84.0-84.9)
	Severe NPDR	0.965(0.963-0.967)	88.3(88.1-88.4)	88.7(88.3-89.0)
	PDR	0.958(0.956-0.960)	89.8(89.2-90.3)	90.7(90.5-90.9)
	Referable DR	0.973(0.971-0.976)	90.1(89.8-90.5)	90.6(90.3-91.0)
NDSP	Non-DR	0.922(0.919-0.924)	83.0(82.6-83.4)	84.2(83.8-84.5)
	Mild NPDR	0.925(0.920-0.930)	85.3(84.6-86.0)	84.5(84.2-84.8)
	Moderate NPDR	0.960(0.954-0.965)	87.1(86.4-87.7)	87.6(86.9-88.2)
	Severe NPDR	0.952(0.947-0.956)	85.0(84.8-85.2)	84.9(84.9-85.0)
	PDR	0.956(0.950-0.961)	87.5(86.9-88.0)	88.1(87.5-88.6)
	Referable DR	0.965(0.962-0.968)	88.5(88.3-88.8)	89.9(89.2-90.7)
EyePACS	Non-DR	0.937(0.934-0.940)	85.9(85.7-86.1)	85.1(84.5-85.6)
	Mild NPDR	0.940(0.937-0.943)	87.1(86.7-87.5)	84.1(83.9-84.3)
	Moderate NPDR	0.937(0.935-0.938)	86.0(86.0-86.0)	85.7(85.4-86.0)
	Severe NPDR	0.953(0.949-0.957)	86.3(85.8-86.7)	86.5(86.1-86.8)
	PDR	0.958(0.955-0.961)	89.5(89.3-89.7)	89.3(88.9-89.7)
	Referable DR	0.948(0.945-0.951)	86.9(86.6-87.2)	86.4(85.8-86.9)

*According to the expert consensus on naming notation from the Chinese Diabetes Society, we changed the notations of the two studies. Shanghai Integrated Diabetic Prevention and Care System Study (SIDPCSS), as referred in the previous version of manuscript, is now called Shanghai Integrated Diabetic Prevention and Care System (Shanghai Integrated Model, SIM). Nationwide Screening for Complications of Diabetes (NSCD), as referred in the previous version of manuscript, is now called

Grading

At a high level, the sections describing grading require deeper consideration and much clarification.

The description for image gradability could be further clarified in lines 154-157: by explaining how the image gradability is classified if macula or optic disc is partially visible in an image. Consider specifying thresholds for extent of visibility required to classify an image as gradable.

Answer: Thank you for this comment. We now specified the thresholds for extent of visibility required to classify an image as gradable. Based on guidelines for image acquisition and interpretation of diabetic retinopathy screening in China (*Chinese Journal of Ophthalmology*, 2017,53, 890–6), the image quality was graded according to standards defined in terms of three quality factors, artifact, clarity, and field definition. We also used this scoring criteria in our previously published study (*Medical Image Analysis*, 2020;61:101654). The score for each factor was listed as below. The total score is equal to the score for clarity plus the score for field definition and minus the score for artifact. The total score less than 12 is considered as ungradable. These scoring criteria were now described (Page 15, Line 317-322) and the following table were added in the Supplementary Table 4.

Supplementary Table 4. Image quality scoring criteria

Type	Image quality specification	Score
Artifact	No artifacts	0
	Artifacts are outside the aortic arch with scope less than 1/4 of the image	1
	Artifacts do not affect the macular area with range less than 1/4	4
	Artifacts cover more than 1/4 but less than 1/2 of the image	6
	Artifacts cover more than 1/2 without fully covering the posterior pole	8
	Cover the entire posterior pole	10
Clarity	Only Level I vascular arch is visible	1
	Level II vascular arch and a small number of lesions are visible	4
	Level III vascular arch and some lesions are visible	6
	Level III vascular arch and most lesions are visible	8
	Level III vascular arch and all lesions are visible	10
Field Definition		
	Do not include the optic disk and fovea	1
	Only contain either optic disk or fovea	4
	Only contain either optic disk or fovea	6
	Only contain either optic disk or fovea	8
	The optic disk and fovea are within 1 pitch diameter of the center	10

Level I vascular arch was defined as the first bifurcations of major trunk veins; Level II vascular arch was defined as the veins deriving from the first bifurcation; Level III vascular arch was defined as the veins deriving from the second bifurcation.

The image grading process has some inconsistencies between the supplement and main paper. Consider revising the “Image reading process” in the supplement. The supplement mentions review by supervisor and lead reader suggesting two overreads but the manuscript mentions one senior grader for overreading when there is disagreement between the first two graders.

Answer: Thank you for raising this issue. The definition of “leader” and “supervisor” is the same in our study. To avoid confusion, the word “supervisor” is used in related context (Page 14, Line 293-294): “The third ophthalmologist who served as the senior supervisor confirmed or corrected when the diagnostic results were contradictory.”

The number of graders is unclear. Table S1 in the supplement describes grader expertise. It is unclear if this describes the pool of graders or the number of graders

per image. Were only two graders involved for the local dataset? If so, the training data is highly biased to these two graders and especially the senior retinal specialist. If not, then was the NSCD dataset graded by all 7 graders for every image?

Answer: Sorry for the mistake in Supplementary Table 1. Here we meant to describe the pool of grader in each study. For the SIM dataset, there was a reading group containing 133 certified ophthalmologists. In all the three cohorts, each image was read by two graders, and a senior supervise would involve if the two results were inconsistent. We have revised it in Supplementary Table 1:

Supplementary Table 1. Summary of the characteristics of the patients in the local and external cohorts

	Datasets		
	SIM*	CNDCS*	NDSP
.....
Grader experience	A reading group containing 133 certified ophthalmologists (≥ 5 years' experience)	A reading group containing 10 certified ophthalmologists (≥ 7 years' experience)	A reading group containing 5 certified ophthalmologists (≥ 5 years' experience)

* According to the expert consensus on naming notation from the Chinses Diabetes Society, we changed the notations of the two studies. Shanghai Integrated Diabetic Prevention and Care System Study (SIDPCSS), as referred in the previous version of manuscript, is now called Shanghai Integrated Diabetic Prevention and Care System (Shanghai Integrated Model, SIM). Nationwide Screening for Complications of Diabetes (NSCD), as referred in the previous version of manuscript, is now called China National Diabetic Complications Study (CNDCS).

Algorithm

More in-depth explanation can be provided to explain how the lesion detection model was developed and evaluated. For example, what similarity measure between test segmentation and ground truth are they using when calculating AUC (binary lesion presence? Overlap of ROIs?) It seems the lesion detection module was only tested on the “local” dataset. If this is true, it is important for the paper to acknowledge this limitation.

Answer: Thank you for the suggestion. We have provided more details on the development of lesion-aware sub-network in the revised Method (Page 16, Line 340-350). The lesion-aware sub-network had two modules: one was the lesion detection module; the other was the lesion segmentation module. The lesion detection module was a binary classifier that predicted whether any kind of lesions exists in a quadrant of the retinal image, as shown in Supplementary Fig. 3. The lesion segmentation module generated mask images to identify different lesions existing in

the retinal images. We used ResNet and Mask-RCNN to form the lesion detection module and lesion segmentation module, respectively. Both ResNet and Mask-RCNN could be divided into two parts: 1) feature extractor, which took images as input and output features, 2) task-specific header, which took the features as input and generated task-specific outputs (i.e., classification or segmentation). Specifically, we chose to use the Mask-RCNN and ResNet with the same feature extractor architecture, so the feature extractor of one network can be easily transferred to another. To train the DeepDR network effectively, we firstly pre-trained a ResNet to form a DR base network. Then we transferred the pre-trained DR base network to the lesion detection module by initializing the feature extractor of lesion detection module using the feature extractor of pre-trained DR base network, followed by fine-tuning the lesion detection module. Then we initialized the feature extractor of lesion segmentation module by reusing the feature extractor of the lesion detection module. The feature extractor layers of the lesion segmentation module were then fixed, and the rest layers of the module were updated during training.

As the answer to your previous comments mentioned above, we have added more details on how the tasks of the lesion detection and segmentation were evaluated in Method (Page 18-19, Line 369-382). Binary lesion detection was tested using AUC, while lesion segmentation was tested using IoU or F-Score. AUC was used to evaluate the lesion detection module as a binary classifier to determine if a quadrant of a retina image contained a certain kind of lesion.

The lesion-aware sub-network was only tested on the local validation dataset, due to the lack of lesion annotations in external cohorts. Further external validation in multiethnic and multicenter cohorts are needed to confirm the robustness of lesion detection and DR grading of the DeepDR system. We have added this issue in the limitation. (Page 13, Line 270-273)

The authors describe multi-task learning as one of the differentiating factors from previous studies. However, it is not clear which hidden layers are shared, whether hard or soft parameters sharing is deployed and why.

Answer: Thank you for the suggestion. We used hard parameter sharing in our study, and all the layers in the feature extractors of ResNet and Mask-RCNN are shared. Firstly, as the datasets available for lesion related learning tasks were relatively limited, using hard parameter sharing was important to reduce the risk of overfitting (*ArXiv Preprint ArXiv:1706.05098*). Secondly, the features extracted by DR base network were suitable to all the DR-related learning tasks (evaluating image quality, lesion analysis and DR grading), so sharing the pre-trained weights facilitate the training of both lesion detection task and lesion segmentation task. Additional

experimental results (shown in Table R3) demonstrated that hard-parameter sharing worked better than soft-parameter sharing for lesion segmentation. We clarified the use of hard parameter sharing and which layers were fixed in the revised Method (Page 16, Line 334-336).

Table R3: Comparison of lesion segmentation results using hard-parameter sharing and soft-parameter sharing.

	Hemorrhage (IoU)	cotton-wool spot (IoU)	Hard exudate (IoU)	Microaneurysm (F-score)
Hard Parameter sharing	0.732	0.707	0.976	0.817
Soft Parameter sharing	0.716	0.703	0.935	0.794

The authors mention the use of transfer learning from one model to another in several places throughout the paper; it is not clear whether tasks are learned one after another, or simultaneously. Consider adding a diagram to visualize the model architecture.

Answer: Thank you for the suggestion. To clarify the transfer learning process, we added a diagram to visualize the DeepDR system in Fig.2. To train the three sub-networks (quality assessment sub-network, lesion-aware sub-network, and DR grading sub-network), Firstly, we pre-trained a ResNet to form DR grading base network. Secondly, we reused the weights of the convolution layers in the pre-trained DR grading base network to initialize the feature extractor of image quality assessment sub-network, and we used hard parameter sharing (by fixing the weights in the feature extractor) for multi-task learning. Thirdly, we transferred the pre-trained DR grading base network to the lesion detection module. The feature extractor of lesion detection module was further reused in lesion segmentation module. Finally, in the DR grading sub-network, we concatenated the features by combining the pre-trained DR grading sub-network with the feature extractor of the lesion segmentation module. Then the weights in the feature extractors of DR grading sub-network were fixed, and the classification header of DR grading sub-network was updated during training. The detailed description of the model architecture was added in Result. (Page 7, Line 130-138, and Fig.2 as shown below)

Fig. 2: Visual diagram of the DeepDR system.

DeepDR system consisted of three sub-networks: image quality assessment sub-network, lesion-aware sub-network, and DR grading sub-network. We first pre-trained the ResNet to form the DR base network (top row). The trained weights of the pre-trained DR base network were then shared in the three different sub-networks of the system, indicated by the red arrow. These three sub-networks took retinal images as input and performed different tasks one by one.

Manuscript Clarity

Line 90: the authors mentioned previous studies have “difficulties identifying early-stage DR.” Is there evidence that supports this statement? Or were previous studies focusing on referable DR rather than early-stage DR?

Answer: Thank you for the comment. We have accordingly revised the related sentences in the manuscript: “However, the importance of identifying early-stage DR should not be neglected.” (Page 5, Line 81)

Line 127: Consider linking Table S1 list of cameras.

Answer: We have linked the sentence with Supplementary Table 1: “In the SIM project, retinal photographs were captured using desktop retinal cameras from Canon, Topcon and ZEISS (Supplementary Table 1).” (Page 14, Line 286-287)

Line 145: Consider adding citation to ICDR.

Answer: We have added citation to ICDR (*Ophthalmology*, 2003,110,1677-82). (Page 15, Line 310)

Line: 180: Maybe reference Table 1 in this section.

Answer: We have linked Table 1 with this section: “The prevalence of DR in the study cohorts is shown in Table 1.” (Page 6, Line 117-118)

Line 187-189: Consider more details for NSCD and NDSP. Were they non-diabetic?

Answer: According to the expert consensus on naming notation from the Chinese Diabetes Society, we changed the notations of the study cohort. Nationwide Screening for Complications of Diabetes (NSCD), as referred in the previous version of manuscript, is now called China National Diabetic Complications Study (CNDCS). We have added more details and revised related description (Page 9, Line 171-175): “The first cohort was the China National Diabetic Complications Study (CNDCS) cohort, comprising 92,672 fundus images from 23,186 diabetic patients and was acquired in 2018. The second cohort was the Nicheng Diabetes Screening Project (NDSP) cohort, comprising 27,948 fundus images from 6,987 elderly subjects over 65 years of age and was acquired in 2018. The prevalence of diabetes was 31.7% in NDSP cohort.”

Line: 267-269: Maybe quantify standard average performance computer.

Answer: We have added the detail on the standard of the computer that is capable to run the DeepDR system in Method: “Any desktop or laptop computer with x86

compatible CPU, 10GB or more free disk space, and at least 8GB main memory is capable to run the DeepDR system. There is no specialized hardware requirement, including GPU or any speed up card, to run the software. A powerful computer with more CPU cores and a GPU will speed up the diagnose procedure significantly, while the diagnose time on a typical laptop (i.e., with Intel I3 processor, no GPU, more than 8G memory) is also acceptable (less than 20 seconds per image).”

Line: 319: Unclear what 30% correlates to.

Answer: Thanks for pointing out this issue. We have revised this sentence: “Based on the feedback information, the artificial intelligence-assisted image quality assessment can reduce the proportion of poor-quality images from 28.7% to 8.2%.” (Page 13, Line 261-263)

Responses to Reviewer #3

Thank you for submitting your paper to this journal.

This is an important topic to cover and one that requires large number of images both for training and for validation; these requirements are met in this report. I have a few questions and remarks for the authors.

1, Abstract: retinal screening does not prevent blindness, it identifies the patients who need to have treatment; the treatment will prevent blindness. Timely referral and treatment resulting from good quality screening means higher likelihood of preventing blindness and so pls rephrase this sentence.

Answer: Thank you for the suggestion. We have rephrased this sentence: “Retinal screening contributes to early detection of diabetic retinopathy (DR) and timely treatment, but is limited by screening efficiency.”

2, Second sentence of the abstract is not accurate: we do have an accurate early detection system (human graders) and they are NOT incapable, that would be an insult to the whole workforce in retinal screening. Please remove or completely re-phrase this sentence. I suspect you are wanting to say that there are not enough appropriately trained healthcare workers to carry out the task of looking at images of all patients with diabetes on a timely manner? However, the way it is presented does not convey this message.

Answer: We apologize for the inappropriate sentence. We totally agree with your comment and have removed the sentence in the abstract. We have added the similar sentences as you pointed out in the Introduction (Page 4, Line 72-74).

3, The abstract has no aim and very little results in it; "high accuracy" is not enough to let the reader know what the DeepDR system can achieve.

Answer: Thank you for the comment. We rewritten the abstract to describe our work in a more comprehensive way:

Abstract

Retinal screening contributes to early detection of diabetic retinopathy (DR) and timely treatment, but is limited by screening efficiency. To solve this problem, we developed a deep learning system, named DeepDR, that can detect early-to-late stages of DR. DeepDR was trained for real-time image quality assessment, lesion detection and DR grading using 492,605 fundus images from 128,630 diabetic patients. Evaluation was performed on a local dataset with 173,778 fundus images from 44,716 patients and three external datasets with a total of 209,322 images. The area under the receiver operating characteristic curves (AUCs) for detecting microaneurysms, cotton-wool spots, hard exudates and hemorrhages were 0.903, 0.938, 0.957 and 0.964, respectively. The classification of DR as mild, moderate, severe and proliferative achieved AUCs of 0.942, 0.957, 0.962 and 0.974, respectively. In external validations, the AUCs for DR grading ranged from 0.921 to 0.963, which further supports the system is efficient for DR screening.

4, What is comparable accuracy to external validation set?

Answer: Thank you for the suggestion. We added the result of external validation set in the abstract: “In external validation datasets, the AUCs for the classification of each level of DR ranged from 0.921 to 0.963.”

5, The last sentence of the abstract is not supported by data or previous evidence; DR screening itself does not produce better healthcare outcomes; acting upon results might.

Answer: Thank you for the suggestion. The sentence was rewritten: “In external validation datasets, the AUCs for the classification of each level of DR ranged from 0.921 to 0.963, which further supports the DeepDR system is efficient for DR screening.”

6, Body of the paper: Diabetic retinopathy (DR) - please use the abbreviation right from the start of the text.

Answer: Thank you for the correction. We have rechecked all the abbreviations to ensure they are placed correctly.

7, China has the largest population of diabetes in the world. - please re-write this sentence.

Answer: Thank you for the correction. The sentence has been rewritten: “China is the country with the largest number of diabetic patients worldwide.” (Page 4, Line 64)

8, It would be important to acknowledge that in countries where systematic DR screening has been carried out, DR is not the leading cause of blindness amongst adults anymore (Iceland and UK), as this concept is important for your arguments later on.

Answer: Thank you for the suggestion. We have added this important concept in the Introduction part, leading to the subsequent scientific question: “In the United Kingdom and Iceland, where systematic national DR screening has been carried out, DR is no longer the leading cause of blindness among working-age adults (*BMJ Open*, 2014,4(2): e004015; *Acta Ophthalmologica Scandinavica*, 1997,75(3):249-54)” (Page 4, Line 59-61).

9, "Few communities have qualified ophthalmologists to carry out DR screening. Patients with severe DR at diagnosis have missed the best treatment opportunity, which leads to poor prognosis or even blindness." - these are two different concepts. Most countries have far too few ophthalmologists and screening is mostly NOT carried out by ophthalmologist, the key issue is to task-shift/task-share so ophthalmologists can do the treatment and not the screening.

DR is NOT treated in early stages and in many cases you can save sight even for PDR patients as long as the sight is good at the start of the treatment, and so this statement is not correct.

Answer: Thank you for the valuable comment. We have corrected this issue in the manuscript. As a cost-effective preventive measure, regular retinal screening is encouraged at the community level. Task shifting is one way the public health community can address this issue head-on so that ophthalmologists can do the treatment but not the screening. Task shifting is the name given by WHO to a process of delegation whereby tasks are moved, where appropriate, to less specialized health workers (*Task shifting: rational redistribution of tasks among health workforce teams: global recommendations and guidelines, WHO*). However, the primary healthcare workers are insufficient to carry out DR screening due to the lack of capacity of assessing retinal images (*British Journal of Ophthalmology* 2017,101,1352-60). Thus,

diagnosis system using deep learning algorithms is required to help DR screening. These descriptions were included in the Introduction (Page 4, Line 69-74).

10, You argue that MAs must be diagnosed so appropriate public health and general diabetes care mechanisms are put in place for the patient to delay the onset of visual loss by better control of diabetes. This is an important concept, but you have to draw it out in the discussion that most of these procedures will not depend on the eyecare providers at this stage of the diabetic eye disease.

Answer: Thank you for your valuable suggestion. This concept has been emphasized in the discussion. The management of diabetes and its complications, including diabetic retinopathy, is systematic work that requires corporation among specialists, general practitioners, and the patients. The identification of patients with mild DR allows health providers other than eyecare providers, such as family physicians, general practitioners, and endocrinologists, to participate in the patient education and management of blood glucose, lipid profiles, blood pressure and other risk factors (*Diabetes Care, 2020, 43(Supplement 1):S135-51*). We have discussed this issue in the revised Discussion (Page 11, Line 217-219).

11, Methods: pls clarify what "grade 2 vascular arch" means, some readers will not be familiar with the terminology and understanding this concept is vital.

Answer: Thank you for the suggestion. We have revised the expression “grade 2 vascular arch” to “level II vascular arch”. Level I vascular arch was defined as the first bifurcations of major trunk veins; Level II vascular arch was defined as the veins deriving from the first bifurcation; Level III vascular arch was defined as the veins deriving from the second bifurcation. We have added the definition of level II vascular arch in the Method (Page 16, Line 338). The definition of Level I/II/III vascular arch have been added to the footnote of Supplementary Table 4.

12, "because it was a community-based cohort rather than a diabetic patients-based cohort". This half a sentence does not belong here and really does not necessarily explain the situation. Either re-word or remove.

Answer: Thank you for your suggestion. We have removed this half of the sentence.

13, What difference did the age-difference between cohorts make?

Was there a difference in duration of diabetes as well?

Answer: The average age in the SIM cohort was 66.22 ± 7.76 years, and the average age was 62.48 ± 4.23 years for CNDCS and 69.11 ± 2.65 years for NDSP. Compared with SIM cohort, CNDCS cohort contained more middle-aged patients and the NDSP

cohort contained more elderly subjects. To avoid potential bias, the external validation set consisting of these two cohorts covered the age range of the development set.

The median duration of diabetes was different among the cohorts and was 7.5 years, 6.4 years, and 5.6 years for SIM, CNDCS, and NDSP, respectively. The duration of diabetes was added in the Supplementary Table 1:

Supplementary Table 1. Summary of the characteristics of the patients in the local and external cohorts

	Datasets		
	SIM	CNDCS	NDSP
.....
Duration of diabetes, years	7.5 (4.1-12.2)	6.4 (6.3-6.5) *	5.6 (2.9-10.2) *
.....

Data are presented as interquartile range. *, vs. SIM, $p < 0.05$.

14, The real-time image feedback is a great feature with far-reaching consequences in the world of screening. If the screeners know immediately if the images are good quality or not, then they will be able to take a photo then and there and the patient does not need to be called back, saving time and effort for all parties. This must be discussed appropriately in the discussion as reducing ungradable rate is one of the key issues in DR screening programmes.

Answer: Thank you for this positive comment. The clinical significance of real-time image quality assessment has been further discussed.

Despite the continuous optimization in digital fundus cameras, aging, experience, lighting, and other non-biological factors resulting from improper operation still results in high percentage of low-quality fundus images, and reacquisition is time-consuming and sometimes impossible (*Can J Ophthalmol*, 2003,38(7),557–68; *Vis Sci*, 2006, 47(3),1120–5). Previous studies on image quality assessment have focused on post-hoc image data processing (*JAMA* 2016,316,:2402-10; *JAMA* 2017.318:2211-23). In this study, a real-time image quality feedback sub-network was implemented in DR screening. Based on the feedback information, the artificial intelligence-assisted image quality assessment can reduce the proportion of poor-quality images from 28.7% to 8.2%. Furthermore, with the improvement of image quality, the diagnostic accuracy was significantly improved, especially for mild DR. This real-time image quality feedback function allows the operators to identify image quality issue immediately and the patients does not need to be called back. It is a promising tool to reduce ungradable rate of the fundus images, thus increasing the efficiency of DR screening. We have discussed this issue in the revised Discussion (Page 13, Line 264-267).

15, Line 250: "especially for microaneurysm". I am not sure that this is entirely appropriate to word this way. All other AI based programmes are good at picking up MAs. Many softwares that grade for "referrable DR" are indeed based on identification of MAs and many over-refer as the criteria is set low so they do not

miss any patients who do require treatment. The same concept of problematic MA detection surfaces from line 293, this is not really raised by others beforehand in the AI field as problematic.

Answer: Thank you for the comment. Problems such as poor image quality, differences in the size of MAs, the closeness of some MAs to the vessels, and the low number of pixels belonging to MAs, which themselves generate an imbalanced data in the learning process have caused difficulty in MA-detection (*Biomed Eng Online, 2019,18(1):67*). According to your comment, we have carefully read the related works about AI-based DR screening system. In the three studies that have been widely cited, MAs and mild NPDR were not specifically discussed (*JAMA, 2016,316(22):2402-10*; *JAMA, 2017,318(22):2211-23*; *Diabetes care, 2018,41(12):2509-16*). In those studies, the deep learning systems were directly trained end-to-end from origin fundus images to the labels of DR grades. During this process, the network might fail to encode the lesion features due to the black-box nature of a deep learning system. In the study of Gulshan et al. (*JAMA, 2016,316(22):2402-10*), the Inception-v3 architecture was trained to make binary prediction of whether the fundus image was referable DR. In the study of Ting et al. (*JAMA, 2017,318(22):2211-23*), the VGG network was trained by presenting the network with batches of labeled training images to recognize referable DR, possible glaucoma, and age-related macular degeneration. In the study of Li et al. (*Diabetes care, 2018,41(12):2509-16*), Inception v3 architecture was trained to binary prediction of whether the fundus image was vision-threatening referable DR. In our study, instead of directly end-to-end training from fundus images to DR grades, we introduced an efficient retinal lesion-aware sub-network based on ResNet to increase the ability of detecting MAs. ResNet avoids the problem of vanishing gradients, which makes it a more sensitive feature extractor for small lesions compared to other network architectures that widely used in previous studies, such as VGG and Inception (*Proceedings of the IEEE conference on computer vision and pattern recognition. 2016: 770-778*). We performed additional experiments evaluating the performance of DR grading using different network architectures in the local validation dataset. The results are shown in Table R4. Our DeepDR system built on ResNet outperformed VGG and Inception in terms of detecting mild NPDR. To describe in a more precise manner, we have revised related sentences in Discussion (Page 11, Line 231-235).

Table R4. Performance of DR grading using different network architectures in the local validation dataset

	DeepDR system built on Resnet			VGG			Inception		
	AUC (95% CI)	Sensitivity (%) (95% CI)	Specificity (%) (95% CI)	AUC (95% CI)	Sensitivity (%) (95% CI)	Specificity (%) (95% CI)	AUC (95% CI)	Sensitivity (%) (95% CI)	Specificity (%) (95% CI)
Non-DR	0.943 (0.939-0.947)	84.4 (83.7-85.0)	89.7 (88.9-90.5)	0.918 (0.912 - 0.924)	83.8 (83.2-84.4)	82.6 (81.9-83.2)	0.936 (0.932 - 0.941)	78.1 (77.9-78.4)	82.2 (81.6-82.8)
Mild NPDR	0.944 (0.940-0.949)	86.3 (85.9-87.8)	85.5 (84.6-86.0)	0.898 (0.892 - 0.905)	81.8 (81.2-82.3)	84.9 (84.6-85.2)	0.881 (0.873 - 0.889)	78.9 (78.8-79.0)	79.9 (79.4-80.3)
Moderate NPDR	0.960 (0.959-0.965)	89.3 (88.4-90.4)	88.0 (87.6-88.8)	0.939 (0.931 - 0.947)	87.2 (86.6-87.7)	87.4 (87.0-87.9)	0.925 (0.922 - 0.927)	84.8 (84.3-85.4)	83.9 (83.2-84.5)
Severe NPDR	0.962 (0.957-0.967)	91.3 (89.8-93.3)	86.3 (86.0-87.3)	0.943 (0.939 - 0.947)	83.8 (83.1-84.5)	86.0 (85.3-86.7)	0.901 (0.896 - 0.905)	87.9 (87.5-88.3)	86.9 (86.1-87.6)
PDR	0.966 (0.964-0.972)	92.3 (90.2-95.4)	91.7 (91.2-92.1)	0.965 (0.957 - 0.972)	90.3 (89.8-90.8)	89.3 (88.7-89.9)	0.963 (0.956 - 0.970)	90.9 (90.4-91.4)	83.8 (83.2-84.5)

16, It would be important to provide details as to what drove the disagreements in terms of what levels of retinopathy/DME were missed and why. It can be just as informative as where there was agreement and it would give clarity to this aspect. How did you ensure that the human grading was good quality? What was the level of agreement there?

Answer: Thank you for the suggestion. It is important to ensure the reading quality of human graders. To ensure the grading quality, we had two certified ophthalmologists (experience ≥ 5 years) to read the same image. The third ophthalmologist who serves as the senior supervisor confirmed or corrected when the diagnostic results were contradictory. The final grading result was dependent on the consistence within these three ophthalmologists. The disagreement rate is 18.9% in the local dataset. The primary cause of the diagnostic divergence was the decision between mild NPDR and non-DR. We have added this issue in the Method (Page 14, Line 298-299).

17, The one topic that is not discussed at all and is one of the major problems in DR screening is detecting change or progression. We know from the UK data that DR progression is indicative of developing sight threatening DR/DME within 4 years and so for an AI software to be able to detect change is imperative. I appreciate that this might be outside of the scope of this paper, but definitely would have to feature in the discussion and the way forward section.

Answer: Thank you for this great advice. AI can detect and quantify retinal features, and that is the basic of detecting DR progression. In addition, AI has the ability to directly compare retina images of the same individual to capture tiny progression of certain kind of retinal lesion. All these technologies could enable us to detect changes even within the same grade of DR. We believe this could be an important and powerful tool in the precise management of DR. We have added these issues in the discussion and the way forward section (Page 12, Line 244-249;Page 13, Line 277-278).

REVIEWER COMMENTS

Reviewer #1 (Remarks to the Author):

We'd like to thank the authors for their thorough response. The revised manuscript addressed many of our concerns. We have summarized the following additional comments that need to be further clarified.

Algorithm

Thank you for clarification on the architecture of the lesion-aware sub-network. We think the paper can benefit from further clarification on this section:

a. Line 85-92: there is significant evidence that deep learning systems can identify novel features in the retinal images which are able to detect or prognosticate outcomes (e.g. <https://www.nature.com/articles/s41551-018-0195-0>). Thus, restricting to detecting known features may actually hamper performance rather than enhance it. The authors should note this caveat when comparing their current work with previous work based on end-to-end training.

b. Multi-task learning is a well-known concept and has been applied to a large variety of learning tasks including those for retinal abnormalities. A quick search reveals many examples: (e.g. https://link.springer.com/chapter/10.1007/978-3-030-32239-7_4, <https://pubmed.ncbi.nlm.nih.gov/30908197/>). The section comparing the present work to previous work should be updated to rephrase the novelty claim around multi-task learning.

c. Line 130-133: while training the initial set of weights for the DR task, how were the weights initialized? The authors do not mention using any pre-trained weights (e.g. those on the ImageNet classification task are common), and this could improve the results and could be one reason why the AUCs reported in this work (even for moderate-or-worse levels) compare unfavorably to previous work.

d. Line 136: the features from the lesion segmentation network are said to be "integrated" with the features extracted in the last DR-detection subnetwork. The authors should elaborate on how this integration was done.

e. Line 138: the authors should clarify datasets and criteria used for early stopping.

f. The concept of using lesion detection as subtasks has been explored before (e.g. <https://www.nature.com/articles/s41746-018-0040-6>). The authors should cite this or related work and note the precedence of this idea in the text.

g. Line 199: as the authors note, the predicted lesion masks could be helpful for making the model more interpretable. However, in the present work, the segmentation output is a separate branch, and the fusion with the final DR-grading task happens before the lesion mask is generated. Given

this, one cannot be certain if the lesion mask output “explains” the final DR grade. This warrants additional investigation before a strong explainability claim can be made. Note that this is in contrast to previous work that uses lesion subnetworks directly on the path to DR grade prediction.

h. Line 324-350: The authors should elaborate on any non-maximum suppression that was deployed as part of Mask-RCNN. This is particularly important since the true-positive and false-positive numbers can be artificially higher as compared to false-negatives if multiple predicted overlapping patches are not sufficiently suppressed, which would in turn affect the F-score.

Data Splits

Thank you for conducting additional experiments to verify the robustness of results. Though the authors explained “operators all received standard training” and “images were read by a centered reading group,” dataset shifts can still come from other factors, for example, camera, disease distribution, etc. Given this, division by districts as the primary data split method raises data uniformity concerns (line 113-115). Consider including and discussing results using random split (Table R1) and multi-fold cross-validation (Table R2).

Ground Truth

There are two images per eye in the dataset and the more severe grade was used as the ground truth for the eye. How are the two images used to generate a DR prediction? Are there separate predictions per image which are then combined, or does the model take both as input to process them together? Authors should elaborate on this.

Results

Thank you for clarifying that hard parameter sharing was used. Consider adding a brief paragraph to discuss results using hard parameter sharing vs. soft parameter sharing (Table R3) in this section, and point readers to the table.

Table 2: We assumed that when an AUC for SevereNPDR is reported, it means AUC is being reported “severe-or-worse” vs “less-than-severe”; and similarly for other levels. Would be good to explicitly clarify this convention.

Table 2: For each AUC number reported in the tables, it would be nice to have the #total and #positives reported alongside.

Reviewer #3 (Remarks to the Author):

Thank you for submitting your answers to the questions.

I still have a few remarks for consideration.

Abstract:

"limited by screening efficiency. To solve this problem," - an AI algorithm will not solve the problem, as image grading is a small part of the screening process, which starts from identifying patients with diabetes, then inviting and photographing them, then the grading process, finally the treatment pathway (or re-screening). Therefore it cannot be claimed that AI software will solve the problem, please re-phrase.

"diabetic patients" - please do not use this wording as such; patients/people with diabetes is the correct term.

"0.921 to 0.963, which further supports the system is efficient for DR screening." - AI software does not screen, it grades the images presented to it and so please do re-consider the wording. Thank you.

"is the earliest stage of DR" - earliest stages of DR when retinal abnormalities are apparent/visible - with electrophysiology and detailed functional testing, there is a possibility for earlier diagnosis, so the wording is not quite correct.

Line 69-74: the concept of task-sharing is an important one, but it is not the "healthcare workers" who are insufficient, it might be their training or where they are placed in the system.

Thank you for answering the questions on methodology. The identification of poor quality images is highlighted and commented upon, and the significance of such a major reduction is appreciated in the text of Discussion; from the clinician point's of view, this is one of most needed additions at primary screening level and will have the largest impact on screening delivery on the community level.

Point-by-point responses (Manuscript No: NCOMMS-19-40862C)

We are thankful for the opportunity to submit our revised manuscript. We thank all the reviewers for their insightful and constructive comments. We have now revised the manuscript to address the reviewers' concerns. Our point-by-point responses are following:

Responses to Reviewer #1

Algorithm

Thank you for clarification on the architecture of the lesion-aware sub-network. We think the paper can benefit from further clarification on this section:

a. Line 85-92: there is significant evidence that deep learning systems can identify novel features in the retinal images which are able to detect or prognosticate outcomes(e.g. <https://www.nature.com/articles/s41551-018-0195-0>).Thus, restricting to detecting known features may actually hamper performance rather than enhance it. The authors should note this caveat when comparing their current work with previous work based on end-to-end training.

Answer: Thank you for the comment. The known features in our study referred to lesion features such as microaneurysms and exudates. Our DR grading network was also trained end-to-end because we finetuned the DR grading network rather than fixed the weight of the feature extractor. Due to the fact that embedding prior knowledge into the end-to-end machine learning algorithms can regulate machine learning models and shrink the search space (*The Journal of Machine Learning Research, 2016,7(1),226-257*), and the ophthalmologists read fundus images based on the presence of lesions, our DR grading network can leverage lesion features as prior knowledge to enhance the performance of DR grading. In addition, our DR grading network did not prevent the DR grading network from learning novel features due to the nature of the end-to-end network. We have added such discussion in Line 266-269: "Due to the fact that embedding prior knowledge into the end-to-end machine learning algorithms can regulate machine learning models and shrink the search space, and the ophthalmologists read fundus images based on the presence of lesions, our DR grading network can leverage lesion features as prior knowledge to enhance the performance of DR grading."

Additionally, we conducted an experiment to compare the performance of our DR grading network with that of other end-to-end networks. As shown in Table R1, fusing lesion features into the DR grading network can enhance rather than hamper the grading performance.

Table R1. DR grading performance comparison of our method with other end-to-end methods.

Local validation		AUC (95% CI)	Sensitivity (95% CI)	Specificity (95% CI)
DeepDR system	Non-DR	0.945 (0.943-0.947)	0.912 (0.911-0.914)	0.825 (0.818-0.831)
	Mild NPDR	0.943 (0.940-0.946)	0.888 (0.877-0.897)	0.839 (0.837-0.841)
	Moderate NPDR	0.955 (0.953-0.957)	0.937 (0.931-0.943)	0.837 (0.834-0.839)
	Severe NPDR	0.960 (0.956-0.964)	0.920 (0.904-0.936)	0.860 (0.858-0.862)
	PDR	0.972 (0.966-0.977)	0.930 (0.898-0.958)	0.895 (0.893-0.897)
	Referable DR	0.973 (0.972-0.974)	0.941 (0.936-0.946)	0.897 (0.895-0.899)
ResNet used in reference 22 (JAMA 2017, 318, 2211-2223)	Non-DR	0.928 (0.924–0.934)	0.761 (0.722-0.799)	0.928 (0.882-0.975)
	Mild NPDR	0.911 (0.906–0.927)	0.771 (0.733-0.810)	0.866 (0.823-0.909)
	Moderate NPDR	0.945 (0.937–0.950)	0.727 (0.691-0.763)	0.911 (0.865-0.956)
	Severe NPDR	0.946 (0.940–0.951)	0.763 (0.725-0.801)	0.977 (0.928-1.026)
	PDR	0.958 (0.953–0.964)	0.766 (0.727-0.804)	0.987 (0.938-1.036)
	Referable DR	0.953 (0.946–0.959)	0.958 (0.910-1.006)	0.894 (0.849-0.939)
Inception-V3 used in reference 21 (JAMA 2016, 316, 2402-2410)	Non-DR	0.924 (0.918–0.930)	0.747 (0.710-0.785)	0.928 (0.882-0.975)
	Mild NPDR	0.916 (0.910–0.921)	0.748 (0.711-0.785)	0.860 (0.817-0.903)
	Moderate NPDR	0.943 (0.937–0.949)	0.730 (0.693-0.766)	0.905 (0.860-0.950)
	Severe NPDR	0.943 (0.937–0.948)	0.750 (0.713-0.788)	0.975 (0.926-1.024)
	PDR	0.956 (0.950–0.963)	0.735 (0.698-0.772)	0.986 (0.937-1.035)

b. Multi-task learning is a well-known concept and has been applied to a large variety of learning tasks including those for retinal abnormalities. A quick search reveals many examples: (e.g. https://link.springer.com/chapter/10.1007/978-3-030-32239-7_4, <https://pubmed.ncbi.nlm.nih.gov/30908197/>). The

section comparing the present work to previous work should be updated to rephrase the novelty claim around multi-task learning.

Answer: Thanks for your constructive comment. Although multi-task learning is widely used, the strategy of our multi-task learning was different from previous methods to adapt to the different purposes and data. Specifically, the DeepDR system was designed as the transfer learning assisted multi-task network, consisting of image quality assessment, lesion detection, lesion segmentation, and DR grading. The main purpose of our multi-task learning was to achieve high accuracy of the DR grading and offer visual hints that help to identify the early stage DR patients. In the DeepDR system, our multi-task learning incorporates with the transfer learning scheme to improve the performance of DR grading task based on the task of lesion detection and segmentation, due to the fact that DR grading relies on the global presences of the DR lesions. Moreover, our multi-task learning method utilized a multi-scale structure, which captured multi-scale features to achieve better lesion segmentation performance.

As mentioned by the reviewer, the main purpose of Xing Wang et al.'s multi-task learning model (*Advances in Neural Information Processing Systems, 19,2006,41–48*) is to recognize the 36 different retinal diseases rather than DR grading. Therefore, their model was first trained to extract the optic disk and macular, and then their multi-task learning networks were trained to classify 36 different retinal diseases. Different from their network, our multi-task network used transfer learning to improve the performance of DR grading task by pretraining on the lesion detection and segmentation tasks.

Clément Payout, et al. designed another multi-task learning architecture (*IEEE Transactions on Medical Imaging, 38,10,2019*) that was trained for three tasks: binary classification of the presence of lesions, segmentation of red lesions, and segmentation of bright lesions. The last two segmentation tasks were conducted concurrently based on a limited number of ground truth labels at the pixel level in their work. Differently, on the other hand, our DeepDR system was first trained to identify the DR lesions containing multi-scale local texture and structures. And our multi-task framework can fuse the features extracted from the detection and segmentation sub-network into our DR grading sub-network to improve grading performance. Therefore, our model can not only achieve the segmentation of lesions but also obtain the DR grading.

As suggested by the reviewer, we have cited those two studies (reference 27,28) and revised related paragraph in the Introduction (Line 86-90): “First, there are few end-to-end and multi-task learning methods can share the multiscale features extracted from convolutional layers for correlated tasks, and further improve the performance of DR grading based on the lesion detection and segmentation, due to the fact that DR grading inherently relies on the global presence and distribution of the

DR lesions”, and Methods (Line 384-389): “The transfer learning assisted multi-task network was developed in our DeepDR architecture to improve the performance of DR grading based on lesion detection and segmentation. Due to the fact that DR grading inherently relies on the global presences of retinal lesions that containing multi-scale local texture and structures, the central feature of our multi-task learning method was designed to extract multi-scale features encoding local textures and structures of retinal lesions, where the transfer learning was used to improve the performance of DR grading task.”

c. Line 130-133: while training the initial set of weights for the DR task, how were the weights initialized? The authors do not mention using any pre-trained weights (e.g. those on the ImageNet classification task are common), and this could improve the results and could be one reason why the AUCs reported in this work (even for moderate-or-worse levels) compare unfavorably to previous work.

Answer: Thank you for the comment. We used ImageNet (*IEEE conference on computer vision and pattern recognition 2009, 248-255*) to pre-train and initialize the DR grading base network. More specifically, the feature extractor layers were pre-trained on ImageNet classification, which contained 1.2 million high-resolution images. Then the pre-trained weights of feature extractor layers were loaded to the corresponding layers in our DR base network. We added a description on the initialization of the network in Line 129-131: "Specifically, a DR base network was first pre-trained on ImageNet classification and then finetuned on our DR grading task using 415,139 retinal images." It might not be fair to directly compare our AUC with that reported in previous works such as *JAMA 2016, 316, 2402-2410*, and *JAMA 2017, 318, 2211-2223*, because the models were trained and tested on different datasets with different characteristics such as fundus cameras, image quality, image grading protocols, populations, etc. In fact, the experimental results have shown that our method outperformed Inception-V3 and ResNet-101 respectively used in the above two previous studies on the SIM dataset, as shown in Table R1. We appreciate your suggestion and will release our test code to encourage more researchers to validate our model on their datasets.

d. Line 136: the features from the lesion segmentation network are said to be "integrated" with the features extracted in the last DR-detection subnetwork. The authors should elaborate on how this integration was done.

Answer: Thank you for the suggestion. We integrated the features by concatenating features. Specifically, if we concatenate two feature vectors $A = (a_1, \dots, a_n)$ and $B = (b_1, \dots, b_m)$, then the concatenated feature is $(a_1, \dots, a_n, b_1, \dots, b_m)^T$. In our work, we concatenate the features from lesion-aware sub-network with the features from the DR grading base network. We updated Line 136-138 by "Furthermore, we

concatenated the lesion features extracted by the segmentation module of the lesion-aware sub-network with features extracted by the DR grading sub-network to enhance grading performance. ". As suggested by the reviewer, we also updated Figure 2 by adding the "concatenate" diagram in the DR grading sub-network, as shown below:

Fig. 2. Visual diagram of the DeepDR system.

e. Line 138: the authors should clarify datasets and criteria used for early stopping.

Answer: We fully agree with the reviewer’s suggestion that the dataset and criteria are crucial for early stopping, and early stopping improves learning efficiency by significantly reducing training time and prevent over-fitting.

We therefore described the dataset and criteria used for early stopping in Line 139-144: "For every task, we randomly split the training dataset into two parts, 80%

of the data were used to train the network and the rest were used for early stopping. The network was tested on early stopping dataset every epoch during training and the performance of the network was recorded. If the area under the receiver operating characteristic curve (AUC) or intersect over union (IoU) increment was less than 0.001 for 5 epochs continuously, we stopped training and selected the best model as the final model."

f. The concept of using lesion detection as subtasks has been explored before (e.g., <https://www.nature.com/articles/s41746-018-0040-6>). The authors should cite this or related work and note the precedence of this idea in the text.

Answer: We agree that lesion detection, as a sub-task, has been used in previous works, such as mentioned by the reviewer. Therefore, we have cited this work at Line 270. in our manuscript.

While both our work and Michael D. Abràmoff et al.'s work (*NPJ digital medicine, 2018,1(1),1-8*) used lesion detection as a subtask, they used a classic learning algorithm rather than an end-to-end deep learning model to fuse the results obtained from lesion detectors, and our DeepDR network, differing from their classic feature fusion, was trained end-to-end with the features extracted from both the lesion-aware sub-network and the original image. In this way, our DR grading sub-network can further exploit the features to minimize the training error, thus improving grading results.

We added a detailed discussion in Line 269-274: "Michael D. Abràmoff et al. used multiple CNNs to detect hemorrhages, exudates, and other lesions, and those detected lesion results were used to classify referable DR by a classic feature fusion model. Differently, our DeepDR network was trained end-to-end with the features extracted from both the lesion-aware sub-network and the original image. In this way, our DR grading sub-network can further exploit the features to minimize the training error, thus improving grading results."

g. Line 199: as the authors note, the predicted lesion masks could be helpful for making the model more interpretable. However, in the present work, the segmentation output is a separate branch, and the fusion with the final DR-grading task happens before the lesion mask is generated. Given this, one cannot be certain if the lesion mask output "explains" the final DR grade. This warrants additional investigation before a strong explainability claim can be made. Note that this is in contrast to previous work that uses lesion subnetworks directly on the path to DR grade prediction.

Answer: Thanks for the comment. We agree that our lesion segmentation output is a separate branch, and the fusion with the final DR-grading task happens before the

lesion mask is generated. As pointed out by the reviewer, "interpretable" in some literature is considered as the extraction of relevant knowledge from a machine-learning model (*arXiv preprint arXiv, 2019,1901,04592*). Our DeepDR machine learning model, like other end-to-end "black box" machine learning models (*JAMA 2016, 316, 2402-2410, JAMA 2017, 318, 2211-2223*), does not have the characteristics of "explainability" in that sense.

In fact, our intention to use "interpretable" in our previous manuscript was that the lesion detection results generated by our DeepDR system can help users to understand the grading result and find lesions in the image. We have therefore updated Line 210-211: "The DeepDR system achieved high sensitivity and specificity in DR grading. Rather than just generating a DR grading, it offers visual hints that help users to identify the presence and location of different lesion types.". We have also conducted an experiment to evaluate the utility of our lesion-aware sub-network as described in the Supplementary Information (Section "Utility evaluation of lesion detection and segmentation by healthcare workers", and Supplementary Table 6):

"We conducted an experiment to evaluate the utility of our lesion-aware sub-network by measuring its effect on the diagnostic accuracy of trained primary healthcare workers from community health service centers, who were not ophthalmologists. We recruited a group of 20 healthcare workers. The experiment followed a within-subjects design to evaluate the grading performance of primary healthcare workers on a sequence of 500 gradable fundus images (100 images for each DR level) randomly chosen from validation set of SIM cohort. All the healthcare workers were shown the same set of 500 fundus images, while the order of the images was randomized. For each fundus image shown, the healthcare workers were asked to grade the image into 0 to 4 DR levels (non-DR, mild NPDR, moderate NPDR, severe NPDR, and PDR). After a healthcare worker made a response, the fundus lesion segmentation generated by the DeepDR system was shown overlaid on the fundus image, and he/she was asked the same question on the DR levels again. Ethics approval was obtained from the Ethics Committee of Shanghai Sixth People's Hospital, informed consent was obtained from graders, and graders were deidentified during analyses.

The DR grading accuracy of the primary healthcare workers was significantly improved with the assistance of lesion segmentation. The results were tested using one-sided, two-sample Wilcoxon signed rank test and are shown in Supplementary Table 6. The sensitivities of all DR grades and the specificity of severe DR were significantly improved with the aid of the DeepDR system."

Supplementary Table 6. The sensitivity and specificity for DR grading by primary healthcare workers with or without help of lesion detection and segmentation

DR levels	Sensitivity			Specificity		
	Unaided	Aided	P	Unaided	Aided	P
Non-DR	0.586 (0.560-0.612)	0.683 (0.679-0.686)	<0.001	0.924 (0.897-0.950)	0.948 (0.937-0.959)	0.504
Mild NPDR	0.589 (0.565-0.613)	0.684 (0.680-0.688)	0.001	0.893 (0.868-0.918)	0.907 (0.896-0.918)	0.362
Moderate NPDR	0.583 (0.554-0.613)	0.680 (0.674-0.687)	0.001	0.914 (0.893-0.936)	0.922 (0.908-0.936)	0.589
Severe NPDR	0.827 (0.822-0.833)	0.843 (0.842-0.844)	0.004	0.873 (0.852-0.894)	0.935 (0.924-0.945)	0.004
PDR	0.827 (0.823-0.831)	0.843 (0.842-0.844)	<0.001	0.903 (0.882-0.924)	0.922 (0.908-0.935)	0.191

h. Line 324-350: The authors should elaborate on any non-maximum suppression that was deployed as part of Mask-RCNN. This is particularly important since the true-positive and false-positive numbers can be artificially higher as compared to false-negatives if multiple predicted overlapping patches are not sufficiently suppressed, which would in turn affect the F-score.

Answer: Thanks for your suggestion. Non-maximum Suppression was used in our lesion segmentation sub-module to select the bounding box with the highest objectiveness score from multiple predicted bounding boxes. Specifically, we first selected the bounding box with the highest objectiveness score, and then compared the IoU of this bounding box with other bounding boxes and removed the bounding boxes with $\text{IoU} > 0.5$. Finally, we moved to the next box with the highest objectiveness score and repeated until all boxes were either removed or selected. As suggested by the reviewer, we added the details on Non-maximum Suppression in Line 373-378: "Non-maximum Suppression was used in our lesion segmentation sub-module to select the bounding box with the highest objectiveness score from multiple predicted bounding boxes. Specifically, we first selected the bounding box with the highest objectiveness score, and then compared the IoU of this bounding box with other bounding boxes and removed the bounding boxes with $\text{IoU} > 0.5$. Finally, we moved to the next box with the highest objectiveness score and repeated until all boxes were either removed or selected."

Data Splits : Thank you for conducting additional experiments to verify the robustness of results. Though the authors explained "operators all received standard training" and "images were read by a centered reading group," dataset shifts can still come from other factors, for example, camera, disease distribution, etc. Given this, division by districts as the primary data split method raises data

uniformity concerns (line 113-115). Consider including and discussing results using random split (Table R1) and multi-fold cross-validation (Table R2).

Answer: Thank you for the suggestion. We agree that random split is the better split method to avoid dataset shifts. Therefore, as suggested by the reviewer, we randomly selected 121,342 subjects (70%) from the SIM cohort into the training dataset, and the remaining 52,004 subjects (30%) were served as the local validation set. The split of SIM cohort was described in Line 113-115 and Fig. 1 as shown below: “Among the 173,346 subjects in the SIM cohort (referred as the local dataset in this study), 121,342 subjects (70%) were randomly selected as the training set, and the remaining 52,004 subjects (30%) were served as the local validation set (Fig. 1).”

We have retrained the DeepDR system and updated related figure (Figure 3) and tables (Table 2, Supplementary Table 2, and Supplementary Table 3) accordingly.

Fig. 1. Data split in the local dataset (SIM cohort) for the training and local validation of the three sub-networks of the DeepDR system.

Ground Truth

There are two images per eye in the dataset and the more severe grade was used as the ground truth for the eye. How are the two images used to generate a DR prediction? Are there separate predictions per image which are then combined, or does the model take both as input to process them together? Authors should elaborate on this.

Answer: Thank you for the comment. In the training process, the input and output of the DR grading network were a single image with a DR grade. For the two images per eye, our DR grading sub-network made separate prediction per image, and then we accepted the more severe DR grade obtained from those images as the grading result

for that eye, which was used to calculate the AUC of DR grades. We revised the sentence in Line 173-175: “For the two images per eye, our DR grading sub-network made separate prediction per image, and then we accepted the more severe DR grade obtained from those images as the grading result for that eye, which was used to calculate the AUC of DR grades.”

Results

Thank you for clarifying that hard parameter sharing was used. Consider adding a brief paragraph to discuss results using hard parameter sharing vs. soft parameter sharing (Table R3) in this section, and point readers to the table.

Answer: Thank you for the suggestion. We have added a brief paragraph in Line 389-394 and Supplementary Table 5 to discuss results using hard parameter sharing vs. soft parameter sharing: "Meanwhile, we used hard parameter sharing in lesion-aware sub-network, and all the layers in the feature extractors of ResNet and Mask-RCNN are shared. Using hard parameter sharing was important to reduce the risk of overfitting (*ArXiv Preprint ArXiv:1706.05098*) due to the limited number of lesion segmentation labels. Besides, sharing the pre-trained weights can facilitate the training of both lesion detection task and lesion segmentation task. Additional experimental results demonstrated that hard-parameter sharing outperformed soft-parameter sharing for lesion segmentation is shown in Supplementary Table 5."

Supplementary Table 5. Comparison of lesion segmentation results using hard-parameter sharing and soft-parameter sharing

	Hemorrhage (IoU)	Cotton-wool spot (IoU)	Hard exudate (IoU)	Microaneurysm (F-score)
Hard Parameter sharing	0.732	0.707	0.976	0.817
Soft Parameter sharing	0.716	0.703	0.935	0.794

Table 2: We assumed that when an AUC for Severe NPDR is reported, it means AUC is being reported "severe-or-worse" vs "less-than-severe"; and similarly for other levels. Would be good to explicitly clarify this convention. For each AUC number reported in the tables, it would be nice to have the #total and #positives reported alongside.

Answer: Thanks for your constructive suggestion. For each level, AUC is reported in a one-vs-all manner, i.e. Mild NPDR means Mild NPDR vs Non-DR, Moderate NPDR, Severe NPDR, and PDR. As suggested by the reviewer, we have explicitly clarified this convention in the table legend of Table 2: "For each level, AUC is

reported in a one-vs-all manner, i.e. Mild NPDR means Mild NPDR vs Non-DR, Moderate NPDR, Severe NPDR, and PDR." In addition, since the number and the prevalence of each DR level among different cohorts in our study were significantly imbalanced as shown in Table 1, positive and negative likelihood, which are independent of disease prevalence (*Modern Epidemiology, 3rd Edition, 2008*), have been added to further elaborate the diagnostic accuracy and their precision, as shown in Table 2 accordingly. Thanks for your kind suggestions.

Responses to Reviewer #3

Abstract:

"limited by screening efficiency. To solve this problem," - an AI algorithm will not solve the problem, as image grading is a small part of the screening process, which starts from identifying patients with diabetes, then inviting and photographing them, then the grading process, finally the treatment pathway (or re-screening). Therefore it cannot be claimed that AI software will solve the problem, please re-phrase.

Answer: Thank you for the suggestion. The sentence was rewritten in the abstract (Line 42-44): "Retinal screening contributes to early detection of diabetic retinopathy (DR) and timely treatment. To facilitate the screening process, we developed a deep learning system, named DeepDR, that can detect early-to-late stages of DR."

"diabetic patients" - please do not use this wording as such; patients/people with diabetes is the correct term.

Answer: Thank you for the correction. We have removed such wording throughout the manuscript and the supplement material.

"0.921 to 0.963, which further supports the system is efficient for DR screening." - AI software does not screen, it grades the images presented to it and so please do re-consider the wording. Thank you.

Answer: Thank you for the suggestion. The sentence was rewritten (Line 50-51): "In external validations, the AUCs for DR grading ranged from 0.916 to 0.970, which further supports the system is efficient for DR grading."

"is the earliest stage of DR" - earliest stages of DR when retinal abnormalities are apparent/visible - with electrophysiology and detailed functional testing, there is a possibility for earlier diagnosis, so the wording is not quite correct.

Answer: Thanks for pointing out the issue. The wording has been corrected according to the website of American Academy of Ophthalmology (<https://www.aaof.org/eye-health/diseases/what-is-diabetic-retinopathy>): “Mild non-proliferative DR (NPDR) is the early stage of DR, which is characterized by the presence of microaneurysms. Proliferative DR (PDR) is the more advanced stage of DR and can result in severe vision loss.” (Line 54-57)

Line 69-74: the concept of task-sharing is an important one, but it is not the "healthcare workers" who are insufficient, it might be their training or where they are placed in the system.

Answer: We totally agree with your comment. The sentence was rewritten in Line 72-74: “Recent evidence has established a role for screening by healthcare workers, given prior training in grading DR (*Br J Ophthalmol* 2017;101:1352–60). However, we still face the issues of insufficiency of their training and where they are placed in the system.”

Thank you for answering the questions on methodology. The identification of poor quality images is highlighted and commented upon, and the significance of such a major reduction is appreciated in the text of discussion; from the clinician point's of view, this is one of most needed additions at primary screening level and will have the largest impact on screening delivery on the community level.

Answer: Thanks for your valuable insight. We have emphasized this aspect in the Introduction accordingly (Line 90-93): “Second, despite being helpful in DR screening, there are few deep learning methods providing on-site image quality assessment with latency compatible with real-time use, which is one of most needed additions at primary DR screening level and will have the impact on screening delivery on the community level.”

REVIEWERS' COMMENTS

Reviewer #1 (Remarks to the Author):

The authors' rebuttal and paper modifications were well thought out and thorough in addressing our concerns. We look forward to reading the published paper.